# THE PERFECT BLEND: REDEFINING RLHF WITH MIXTURE OF JUDGES

## ABSTRACT

Reinforcement learning from human feedback (RLHF) has become the leading approach for fine-tuning large language models (LLM). However, RLHF has limitations in multi-task learning (MTL) due to challenges of reward hacking and extreme multi-objective optimization (i.e., trade-off of multiple and/or sometimes conflicting objectives). Applying RLHF for MTL currently requires careful tuning of the weights for reward model and data combinations. This is often done via human intuition and does not generalize. In this work, we introduce a novel post-training paradigm which we called Constrained Generative Policy Optimization (CGPO). The core of CGPO is Mixture of Judges (MoJ) with cost-efficient constrained policy optimizers, which can identify the perfect blend in RLHF in a principled manner. It shows strong empirical results, does not require extensive hyper-parameter tuning, and is plug-and-play in common post-training pipelines. Together, this can detect and mitigate reward hacking behaviors while reaching a pareto-optimal point across an extremely large number of objectives.

Our results show that CGPO consistently outperforms other commonly used SoTA RLHF algorithms (such as PPO and DPO) on a wide range of tasks – general chat, STEM questions, instruction following, math, coding and knowledge. In particular, CGPO improves over PPO by 7.4% in AlpacaEval-2 (general chat), 12.5% in Arena-Hard (STEM & reasoning), 2% in IFEval (Instruction Following), 2% in both MATH and GSM8K (Math & reasoning), 5% in HumanEval (Coding), and 2% in the ARC challenge (Knowledge). We also observe that PPO is susceptible to severe reward hacking behaviors (it exhibits severe regression in popular coding benchmarks) which can be addressed by CGPO. CGPO represents a breakthrough in RLHF, simultaneously addressing reward-hacking and extreme multi-objective optimization, and thereby advancing the state-of-the-art in aligning general-purpose LLMs.

## 1 INTRODUCTION

The emergence of general-purpose Large Language Models (LLMs) has significantly transformed the landscape of natural language processing, demonstrating exceptional capabilities across various expert-level domains (Achiam et al., 2023; Brown et al., 2020; Touvron et al., 2023; Anthropic, 2023; Team et al., 2023; Meta, 2024; Tunstall et al., 2023; Zhu et al., 2023). These models are characterized by their extensive parameterization, enabling them to handle a wide array of tasks using a unified parameter set (Zhao et al., 2018; Liu et al., 2019b;a). Central to this versatility is multi-task learning (MTL) (Caruana, 1997; Crawshaw, 2020), a strategy that involves training a single model on multiple tasks simultaneously. This approach fosters the development of shared representations, which enhances the model's ability to generalize better than those trained on isolated tasks. Although prior studies on MTL have concentrated on the integration and processing of multi-task data during both pre-training and fine-tuning stages (Raffel et al., 2020; Liu et al., 2023; Aghajanyan et al., 2021; Aribandi et al., 2021), the application of the primary LLM alignment method, Reinforcement Learning with Human Preference (RLHF) (Ouyang et al., 2022; Ziegler et al., 2019; Zheng et al., 2023b), has not been thoroughly explored within the MTL context. In previous studies, the implementation of RLHF for multi-task post-training has typically involved a linear combination of multiple reward models within the standard RLHF framework (Ramamurthy et al., 2022; Glaese et al., 2022; Yuan et al., 2023; Bakker et al., 2022; Wu et al., 2024; Li et al., 2020). Each reward model is crafted

using preference data to mirror the distinct alignment preferences of different tasks. Researchers often experiment with various reward weightings to identify a Pareto front that depicts the optimal performance of the LLM across diverse tasks (Rame et al., 2024). However, this approach is limited by two significant challenges:

**Vulnerability to Reward Hacking:** The optimization of a preference-based reward model is susceptible to reward hacking, as the reward model is an imperfect proxy of human preferences (Gao et al., 2023; Jin et al., 2023; Skalse et al., 2022). Studies indicate that excessive optimization of a reward model can lead to misalignment with actual human preferences (Gao et al., 2023; Moskovitz et al., 2023; Stiennon et al., 2020; Rafailov et al., 2024a). This issue becomes more pronounced in a multi-task setting, where each reward model may have its own unique flaws. Implementing a uniform early stopping point in the RLHF optimization process to minimize reward hacking effects is impractical and can lead to degraded performance across tasks (Moskovitz et al., 2023). This highlights the need for a more tailored approach to compensate for the weaknesses of each reward model and to manage the optimization of reward models for each task in complex, multi-task environments.

**Contradictory Goals:** Different tasks often have conflicting objectives (Rame et al., 2024). Even if the prompt spaces for these tasks do not overlap, using a linear combination of reward models can lead to compromises in goal metrics. For example, the typical strategy of LLM post-training involves maximizing the helpfulness reward for safe prompts and maximizing the harmfulness reward for unsafe prompts (Bai et al., 2022). Although achieving global optimality for both tasks is possible if the LLM's capacity is sufficiently large (Iyer et al., 2022), employing a linear combination of helpfulness and harmfulness rewards inevitably results in reduced gains for both metrics. This occurs because each task partially sacrifices its own RLHF optimization progress to accommodate a contradictory metric, thereby diminishing the effectiveness of both.

To address these challenges, we developed an innovative framework called Constrained Generative Policy Optimization (CGPO). In response to the issue of reward hacking in RLHF, we introduce two types of judges: rule-based and LLM-based. These judges collaborate to identify any reward hacking patterns during the LLM's online generation phase. Based on their evaluations, we implement a constrained RLHF method to update the LLM model. This method is designed to maximize the likelihood of generating outputs that adhere to all constraints and achieve high reward values, while minimizing outputs that breach constraints and have low reward values. To support the constrained policy optimization update in the large-scale LLM setting, which is complicated even in traditional small-scale RL scenarios, we have developed three new primary-type constraint RLHF optimizers. These optimizers are designed to operate independently of the dual-variable update, which is often a critical component in conventional primal-dual constrained RL algorithms. This independence simplifies the optimizers and enhances their scalability, making them more effective for managing large-scale LLM post-training. To effectively optimizing objectives of various tasks, which may be

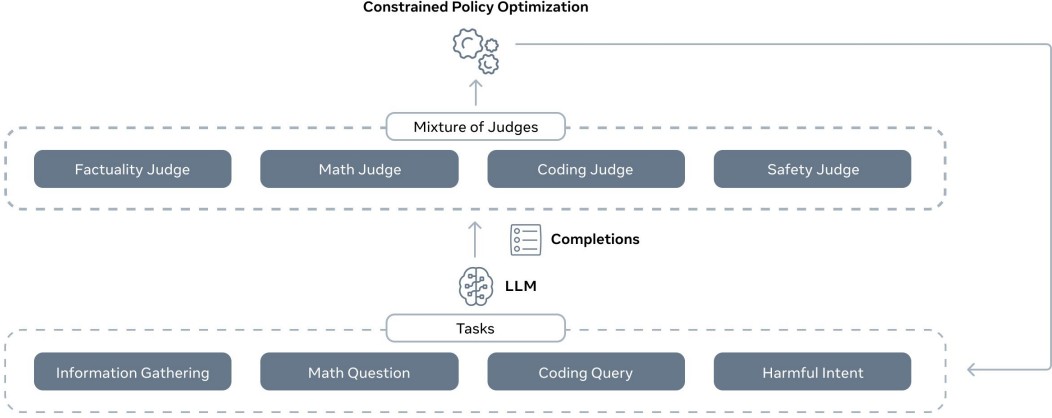

Figure 1: In CGPO, a customized MoJs is applied to each task to evaluate model generations, and the model is updated through our proposed constrained policy optimizer.

contradictory, we propose a novel design in CGPO for managing multi-task post-training. In this design, prompts are segregated by task, and a customized policy optimization strategy is applied

to each set of prompts. This strategy includes a tailored MoJs, reward model, and hyperparameter setup for the constrained RLHF optimizer. By optimizing each task independently, our approach avoids compromises due to conflicting goals from other tasks, a common issue in previous works that used a linear combined reward model. Furthermore, our design addresses the reward hacking issue and optimizes objectives for each task in a fine-grained manner, resulting in a better Pareto frontier than previous methods that enforced uniform treatment across all tasks. See Figure 1 for an overview of our CGPO pipeline.

We summarize our contributions as follows:

- We developed a new strategy to address the issues of reward hacking through an innovative primal-type constrained RL method. To implement this method, we have developed three new constrained RLHF optimizers: Calibrated-Regularized Policy Gradient (CRPG), Constrained Online Direct Preference Optimization (CODPO), and Constraint-Regularized Reward Ranking Finetuning (CRRAFT). All proposed methods are scalable and easy to implement.

- To support the implementation of the constrained RL optimizers, we developed two types of judges: the rule-based judge and the LLM-based judge. These judges are designed to effectively assess whether an LLM generation violates constraints in a broad spectrum of LLM tasks.

- We introduced a new multi-objective RLHF treatment strategy within CGPO, where each task is managed individually with a customized optimization setting, including reward models, mixture of judges, and optimizer hyperparameters. This pioneering design, the first in the multi-task RLHF field, significantly enhances the Pareto frontier across multiple metrics.

- We demonstrate the effectiveness of CGPO in a challenging multi-task post-training environment with five tasks: general chat, instruction following, math and coding reasoning, engagement intent, and safety, despite potentially contradictory goals across tasks. Notably, by primarily utilizing open-source data and the Llama3.0 70b pre-trained model, our research demonstrates that, in comparison to the baseline RLHF methods such as PPO Schulman et al. (2017) and DPO Rafailov et al. (2024b), our approach—when combined with the CRPG and CRRAFT optimizers—consistently outperforms these baselines across all benchmarks and tasks. Specifically

  1. CRPG optimizers achieve the highest performance in terms of MATH, GSM8K, HumanEval, MBPP, ARC Challenge, and false refusal ratio. CRRAFT optimizer achieves the highest performance in AlpacaEval-2, Arena-Hard, and TruthfulQA.
  2. PPO experiences a significant drop in the 0-shot coding benchmarks (HumanEval and MBPP) after exceeding certain training steps, indicating the occurrence of severe reward hacking issues. In contrast, CGPO not only avoids such regression but also consistently improves those benchmarks during training, demonstrating the extraordinary capability of MoJs in preventing reward hacking issues.

## 2 Preliminaries

In the RLHF finetuning phase, we typically formulate a Markov Decision Process (MDP) as follows: each prompt is considered as the state $s$, and the entire response is the action $a = [a_0, a_1, \cdots, a_{T-1}]$, where $a_i \in A$ represents the token at position $i$ and $A$ is the vocabulary set. An LLM policy is defined as $\pi(a_t | a_{t-1}, a_{t-2}, \cdots, a_0, s)$, which represents a distribution over $A$ at time step $t$, conditioned on all previous response history before $t$ and prompt: $\{a_{t-1}, a_{t-2}, \cdots, a_0, s\}$.

### 2.1 Reward Model Training

RLHF starts by finetuing a pre-trained LLM using supervised learning on high-quality dataset relevant to the downstream target task(s) to obtain $\pi_{\text{SFT}}$. After the supervised fine-tuning (SFT) stage, we need to develop a reward model (RM) to assess the quality of an LLM's output. This will enable us to utilize exploration-based online RL alignment method. We typically use the pairwise preference reward model (Stiennon et al., 2020) with Bradley-Terry (BT) formulation (Bradley & Terry, 1952). To learn a parameterized reward model $r_\phi(s, a)$, given a pre-collected preference-pair dataset $\mathcal{D} = \{s_i, a_{w,i}, a_{l,i}\}_{i=1}^N$, where $a_{w,i}$ and $a_{l,i}$ denote the preferred and less preferred generations respectively, we can learn $r_\phi$ by framing the problem as a binary classification and solving the subsequent

problem (Ouyang et al., 2022; Touvron et al., 2023; Meta, 2024):

$$\min_{\phi} \mathcal{L}_{pair}(r_{\phi}, \mathcal{D}_{pair}) = -\mathbb{E}_{\mathcal{D}_{pair}}\left[\log \sigma(r_{\phi}(s, a_p) - r_{\phi}(s, a_n))\right]. \tag{1}$$

## 2.2 RL Finetuning

Given a LLM policy $\pi_w$ with parameter $w$, a reward model $r_{\phi}(a, s)$ and a prompt set $\mathcal{D}_p = \{s_i\}_i^M$, we aim to optimize the policy by maximizing the following RL objective (Ouyang et al., 2022; Achiam et al., 2023; Touvron et al., 2023):

$$\max_{w} \quad \mathbb{E}_{s\sim\mathcal{D}_p, a\sim\pi_w}\left[r_{\phi}(s, a)\right]. \tag{2}$$

When solving the problem in eq. (2) we typically initialize $\pi_w$ with SFT policy $\pi_{\text{SFT}}$ instead of starting from scratch. In previous works a number of online RL method such as proximal policy optimization (PPO) (Schulman et al., 2017), reward ranking (RAFT) (Dong et al., 2023) and REIN-FORCE (Williams, 1992) has been utilized to solve eq. (2).

## 3 Constraint Generative Policy Optimization

In this section, we first explore how to implement the CGPO framework for single objective opti-mizaiton in the single task setting using MoJs, as detailed in Section 3.1. Subsequently, we discuss the implementation of CGPO to manage scenarios involving multiple objectives in Section 3.2 for multi-task learning.

## 3.1 CGPO in Single Task with Single Objective

The primary design of CGPO is to integrate multiple constraints to mitigate the issue of reward hacking, which arises from the limited capabilities of reward models. Specifically, in addition to optimizing the accumulated reward model value as shown in eq. (2), we also ensure that the model generation meets several constraints. For example, in general chat tasks with prompts that are free of harmful intent. We require model generations to consistently respond to user queries. This is crucial because there are instances where the model may refuse to answer, and the reward model might er-roneously assign high values to such non-responsive generations. In these cases, purely maximizing the reward model could impair the model's helpfulness and lead to an overly conservative tendency. By introducing these constraints based on our prior knowledge about the weaknesses of each reward model, we can avoid critical reward hacking patterns effectively.

We denote the set of constraints that the LLM generations need to satisfy as $\{C_1, C_2, \dots, C_M\}$ and the state-action set that satisfies constraint $C_k$ as $\Sigma_k$, i.e., $\Sigma_k = \{(s, a) \in \mathcal{S} \times \mathcal{A} \text{ and } (s, a) \text{ satisfies requirement of } C_k\}$. We define the feasible region as the state-action set that satisfies all constraints as $\Sigma = \Sigma_1 \cap \Sigma_2 \cap \dots \cap \Sigma_M$. In the single task setting, CGPO solves the following constrained problem (Ying et al., 2022; Zhang et al., 2024; Xu et al., 2021)

$$\max_{w} \quad \mathbb{E}_{s\sim\mathcal{D}_p, a\sim\pi_w}\left[r_{\phi}(s, a)\right]$$
$$\text{s.t.} \quad \text{Prob}_{s\sim\mathcal{D}_p, a\sim\pi_w}((s, a) \in \Sigma) \geq 1,$$
$$\text{KL}_{s\sim\mathcal{D}_p}(\pi_w|\pi_{\text{ref}}) \leq \text{KL}_{\max}, \tag{3}$$

where $\pi_{\text{ref}}$ is the initialization model and $\text{KL}_{\max}$ is the threshold of KL-divergence.

The high-level framework of CGPO in the multiple-constraints and single-objective setting is illus-trated in Algorithm 1. At each iteration, we sample a minibatch from the prompt set $D$, and then apply the current LLM policy to generate $K$ responses ($1 \leq K$) for each prompt. Subsequently, we apply all judges $J = \{J_h\}_{h=1}^M$ to generated sample to evaluate whether a generation violates any constraint, where $J_h(s, a) = 1$ if $(s, a)$ satisfies the $h$-th constraint, and $J_h(s, a) = 0$ otherwise. We label a generation $a_{t,i}^k$ as "violated" if it fails any one of the constraint judgments, and "satisfied" otherwise. The judge is a module for evaluating the constraint satisfaction conditions, which could be a rule-based script or an LLM classifier. This module can address a wide range of constrained problems in the LLM post-tuning scenario. We will discuss this in detail in Section 3.1.1.

After that, we split the generations into "Positive" and "Negative" groups, depending on the constraint satisfaction label. We then apply a constrained RLHF optimizer to update the policy with these two groups of samples (see line 9 in Algorithm 1). In our work, we propose three new constrained RLHF optimizers to efficiently solve the multi-constraint problem in the LLM setting. For Option I in Algorithm 1, we develop a policy gradient approach named Calibrated Regularized Policy Gradient (CRPG) and an online direct preference-based approach named Constrained Online DPO, and for Option II in Algorithm 1, we develop a reward ranking-based approach named Constraint-Regularized Reward Ranking Fine-tuning (CRRAFT).

- **Calibrated Regularized Policy Gradient:** CRPG is a constrained policy gradient method. It incorporates a novel calibration strategy that leverages preference-based reward modeling, along with a new constraint-rectified reward shaping technique. Those two techniques work together to optimize the reward while ensuring compliance with all constraints. Additionally, CRPG introduces a new KL-regularization approach that not only penalizes generations with significant deviation but also strictly bound the KL divergence of final policy.

- **Constraint-Regularized Reward Ranking Fine-tuning:** CRRAFT is a reward ranking-based approach Dong et al. (2023). It adopts a novel ranking strategy that promotes only those generations which achieve high reward values and satisfy all constraints. Additionally, this strategy ensures that the KL divergence of the final policy is strictly bounded.

- **Constrained Online DPO:** CODPO adapts the DPO update to the on-policy optimization setting, in which generations that achieve high reward values and satisfy all constraints are promoted, whereas generations that yield low reward values and violate any constraints are demoted.

Please refer to Appendix B for detail about these three constrained policy optimizers.

---

**Algorithm 1** CGPO($D, \pi_{w_0}, J, B, R, O, T$) in single task with multi-constraints

---

1: **Input:** prompt set $D = \{s_{t,i}\}_{i=1}^{N}$, LLM starting policy $\pi_{w_0}$, constraint judge set $J = \{J_h\}_{h=1}^{M}$, batchsize $B$, reward model $R$, iteration number $T$, constriained RLHF optimizer $O$.
2: **for** $t = 0, 1, ..., T$ **do**
3:     Prompt sampling: $\{s_{t,i}\}_{i=1}^{B} \sim D$
4:     Response generation: $\{a_{t,i}^{k}\}_{k=1}^{K} \sim \pi_{w_t}(\cdot|s_{t,i})$ for $1 \leq i \leq B$
5:     Constraint judgement: $y_{t,i}^{k} = \wedge_{h=1}^{M} J_h(s_{t,i}, a_{t,i}^{k})$ for $1 \leq i \leq B$ and $1 \leq k \leq K$
6:     Split sample set:
7:         Positive samples: $X_t^{+} = \{(s_{t,i}, a_{t,i}^{k})$ for $1 \leq i \leq n, \ 1 \leq k \leq K$ where $y_{t,i} = 1\}$
8:         Negative samples: $X_t^{-} = \{(s_{t,i}, a_{t,i}^{k})$ for $1 \leq i \leq n, \ 1 \leq k \leq K$ where $y_{t,i} = 0\}$
9:     Update $\pi_{w_t} \rightarrow \pi_{w_{t+1}}$ for policy optimization with optimizer $O$ and reward model $R$:
10:       **[Option I]:** maximize likelihood of $X_t^{+}$ and minimize likelihood of $X_t^{-}$
11:       **[Option II]:** maximize likelihood of $X_t^{+}$
12: **end for**

---

Intuitively, with either the Option I or Option II updating strategy, CGPO encourages the policy to explore regions that satisfy all constraints to maximize the expected reward model value. Note that CGPO is a primal-type constraint policy optimization approach, which differs from the standard primal-dual approach adopted in the traditional constrained RL field. CGPO does not involve co-optimizing the dual variable, thus avoiding the drawbacks of extensive hyperparameter tuning issues associated with the primal-dual approach. Due to this reason, CGPO is user-friendly even with multiple different types of constraints, making it well-suited for the LLM post-tuning scenario.

### 3.1.1 JUDGES IN CGPO

The key step in implementing multi-constraint CGPO optimizers is to determine whether a generation $(s, a)$ satisfies a constraint or not. This determination allows us to split generated samples into positive ($X_t^{+}$) and negative ($X_t^{-}$) groups given the label $y$ predicted by each constraint judge $J_h$, i.e.,

$$J_h(s, a) = y \in \{0, 1\}, \ \text{where} \ 1 \leq h \leq M,$$

and then apply our customized constraint RLHF optimizers based on that classification. In CGPO, we have developed and integrated the following two types of constraint judge modules to assess whether a generation satisfies a constraint:

- **Rule-based constraint judge module:** This module employs a rule-based approach (such as string-matching and code execution) to ascertain whether the generation strictly adheres to pre-defined regulations (Li et al., 2024a). It is particularly effective for constraints related to precise instruction following, where the generation must meet exact requirements such as length, number of paragraphs, and keyword inclusion (Zhou et al., 2023; Hendrycks et al., 2021b; Cobbe et al., 2021). It can also handle reasoning tasks, such as math problems and code generation.

- **LLM-based constraint judge module.** This module functions as an LLM generator. In most cases, the generation is formatted according to a template before being sent to the judge module. These modules not only provide access to the constraint satisfaction condition but also offer reasoning behind the judgement construction. Due to this property, they are typically capable of handling more challenging constraint evaluation tasks such as safety violation, reference-based factuality verification, and false refusal patterns. The model could either be a compact LLM fine-tuned with domain-specific data (Inan et al., 2023; Bai et al., 2022) or a powerful, large LLM without task-specific fine-tuning (Yuan et al., 2024b; Zheng et al., 2024).

A detailed information of these two types of judges can be found in Appendix C.4.

## 3.2 CGPO in Multi-Taks with Multi-Objectives

In the multi-tasks environment, CGPO utilizes customized combinations of "reward models + MoJs + optimizers" to provide alignment guidance tailored to each task. This approach is designed to better accommodate the specific nature of each problem, thereby enable CGPO to have better chance to achieve optimal alignment outcomes. Figure 2 provides an end-to-end illustration of how the

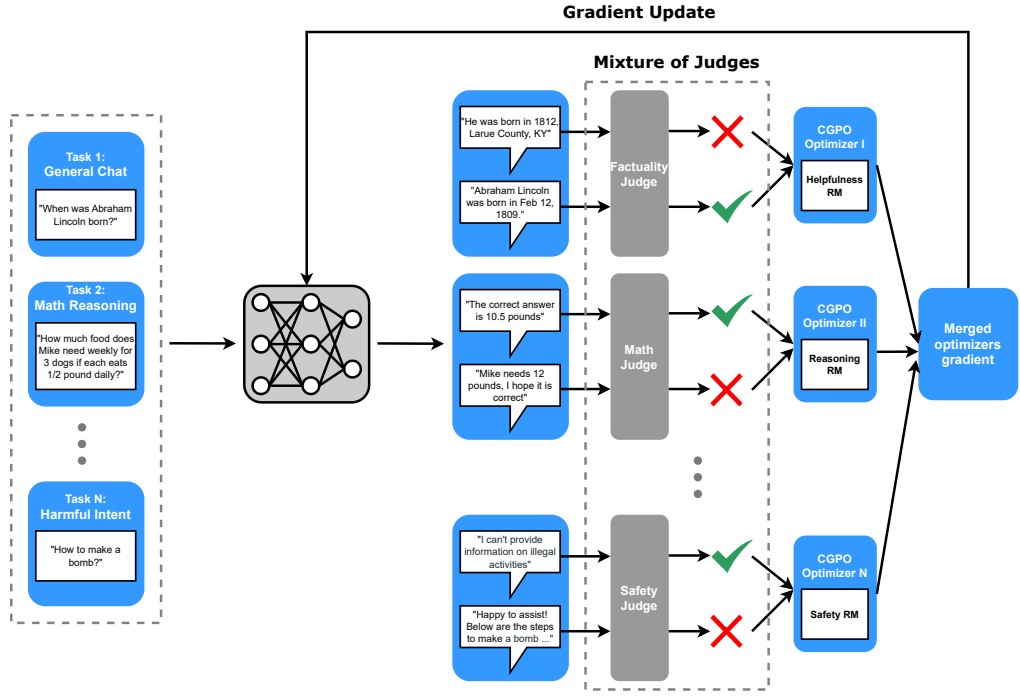

Figure 2: CGPO in a multi-tasks setting. The RM, MoJs, and optimization setup are uniquely tailored to the specific characteristics of each task. This customization ensures the most effective and targeted approach for achieving optimal performance across all tasks, even those with potentially contradictory goals.

CGPO pipeline functions in the multi-tasks setting. The entire CGPO pipeline has the following two core components: multi-objective reward modeling and multi-experts alignment.

**Multi-Objective Reward Modelling.** Unlike the approach adopted in previous RLHF pipelines in multi-objective scenarios, which applies the same linear combined reward model to all prompts in the prompt set $D$, CGPO first classifies the prompt set $D$ into distinct, non-overlapping categories based on the nature of the prompts, i.e., $D = \{D_1, D_2, \ldots, D_L\}$. Each prompt set $D_l \in D$ is referred to as a task. For example, prompts with harmful intent, which could potentially lead LLM to generate unsafe responses, are grouped into a class labeled "harmful intent". Conversely, prompts without unsafe intent, primarily focused on information gathering and casual conversation, are grouped into a class labeled "general chat". This categorization can be performed during the data collection phase or by prompting an LLM to carry out the categorization given the definitions of different classes. Subsequently, with a collection of trained reward models denoted as $\{r_{\phi,1}, r_{\phi,2}, \ldots, r_{\phi,V}\}$, we tailor the specific reward model to be applied for each task $D_l$. This customization guarantees that each prompt class $D_l$ benefits from the most appropriate guidance provided by the corresponding reward model. Note that the number of reward models, denoted by $V$, is less than or equal to the number of tasks, $L$, meaning a single reward model can be utilized across multiple tasks. The major advantage of segregating the reward models for different tasks is to exclude irrelevant or contradictory objectives, thus enabling each task to focus solely on optimizing its own goal metrics without interference from other objectives.

**Multi-Expert Alignment.** The concept of multi-expert alignment involves applying customized MoJs, reward model and policy optimization setups for each task.

After the policy model generates online samples for each task, we employ a mixture of task-specific judges to identify generations that do not meet predefined standards. It is crucial to emphasize that the selection of judges are uniquely tailored for each task, reflecting the particular shortcomings of each reward model and our established performance criteria for LLMs in these tasks. For instance, in the "general chat" task, we employ LLM-based judges for false refusal and factuality to enhance responsiveness and ensure honesty. In "reasoning" tasks, we implement a rule-based math/coding constraint judge to guarantee correctness and accuracy.

Based on the status of constraint satisfaction across generations and a customized reward model, we implement an RLHF policy optimizer with a specifically tailored hyperparameter setup to align each task effectively. This method deviates from the conventional RLHF pipeline, which generally employs a uniform optimizer setup for task alignment. For tasks that have precise judges and require extensive exploration to derive the correct response, such as instruction following, math, and coding, we apply a lenient KL threshold and allow a higher number of generations per prompt. In contrast, for tasks where precise judges are lacking and extensive exploration is less critical, such as "general chat," we opt for a stricter KL threshold and a reduced number of generations per prompt.

---

**Algorithm 2** CGPO($\{D_l\}_{l=1}^L, \pi_{w_0}, \{J_l\}_{l=1}^L, \{B_l\}_{l=1}^L, \{R_l\}_{l=1}^L, \{O_l\}_{l=1}^L, T$) in multi-tasks with multi-constraints & multi-objectives

1: **Input:** Multi-tasks prompt set $\{D_l\}_{l=1}^L$, LLM starting policy $\pi_{w_0}$, judges sets $\{J_l\}_{l=1}^L$, multi-tasks batchsizes $\{B_l\}_{l=1}^L$, reward model sets $\{R_l\}_{l=1}^L$, multi-tasks weights $\{\rho_l\}_{l=1}^L$, multi-tasks optimizers $\{O_l\}_{l=1}^L$, iteration number $T$.
2: **for** $t = 0, 1, \cdots, T$ **do**
3:    **for** $l = 0, 1, \cdots, L$ **do**
4:       Obtain gradient $\tilde{g}_l(\pi_{w_t})$ for $l$-th task via CGPO($D_l, \pi_{w_t}, J_l, B_l, R_l, O_l, 1$) in Algorithm 1
5:    **end for**
6:    Update with multi-tasks gradient accumulation $w_{t+1} = w_t + \alpha_t \cdot \sum_{l=1}^L \rho_l \cdot \tilde{g}_l(\pi_{w_t})$,
7: **end for**

---

The high-level framework of CGPO in the multiple-constraint and multiple-objective setting is illustrated in Algorithm 2. Specifically, at each iteration $t$, we process each individual task to compute the updated gradient $\tilde{g}_l(\pi_{w_t})$. This computation is based on the task-specific prompt set $D_l$, reward model $R_l$, mixture of judges $J_l$, batch size $B_l$, and optimizer $O_l$, following the steps outlined in Algorithm 1. Subsequently, we accumulate the gradients across all tasks and combine them with our predefined task weights $\{\rho_l\}_{l=1}^L$, which are then used to update our model parameters.

## 4 EXPERIMENTS

In this section, we outline the specifics of our experimental setup designed for multi-task alignment under conditions of extreme multi-constraints and multiple objectives. Specifically, we focus on fine-tuning a LLM to achieve alignment across the following five tasks: general chat, instruction following, math/code reasoning, engagement intent and harmful intent (see Appendix C.1 for detail).

In our experiment, we utilized the Llama 3.0 70B pretrained model as our base model. In the SFT stage, we utilize a combination of open-source and synthetic finetuning datasets to enhance model's performance across five specific tasks. We then use both open-source and synthetic preference datasets to train three RMs: Helpfulness RM, Engagement RM, and Safety RM, each designed to capture different aspects of alignment. Additionally, we have developed five judges to support the CGPO training in this context: False Refusal Judge, Precise Instruction Following Judge, Math & Coding Judge, Factuality Judge, and Safety Judge. For detailed information on the SFT and RM training recipes, as well as the development of these judges, please refer to Appendices C.2 to C.4.

We evaluate the capability of models trained with different algorithms using the following benchmarks: AlpacaEval-2, Arena-Hard, IFEval, MATH, GSM8K, MBPP, HumanEval, MMLU, ARC, and TruthfulQA. Additionally, we have developed new benchmarks to assess engagement intent, safety violation rate, and false refusal rate. Please refer to Appendix D for detail.

### 4.1 CGPO TRAINING SETUP

In this section, we will show how we implment the CGPO in the RLHF finetuning stage.

**RLHF warm-up.** Unlike previous studies Ouyang et al. (2022), which directly employ the SFT model as the initial point for RLHF, our approach introduces a "warm-up" phase. This phase begins with a model that has undergone preliminary fine-tuning through a few steps of DPO, starting from the SFT model. The rationale behind this strategy is that initiating online RLHF directly from the SFT model and performing policy optimization with the RM may not be able to explicitly exploit the high-quality preference data. By initiating RLHF with a model already refined by DPO to a certain degree, we can fully harness the advantages of the preference dataset, thereby providing a better starting point for RLHF. In our experiment, we utilize all preference data from RM training to facilitate the training for warm-up DPO. The benefit of RLHF warm-up is discussed in Appendix E.

**RLHF Training recipe:** We begin the RLHF finetuning process using the warm-up model. Table 1 shows the customized treatment (RM+MoJs) we applied for each task. The prompt set of each tasks in CGPO training is provided in Appendices C.5 and C.6.

| Tasks | General Chat | Instruction Following | Math/Coding Reasoning | Engagement Intent | Harmful Intent |
|---|:---:|:---:|:---:|:---:|:---:|
| **Helpfulness RM** | ✓ | ✓ | ✓ | | |
| **Engagement RM** | | | | ✓ | |
| **Safety RM** | | | | | ✓ |
| **False refusal Judge** | ✓ | | | ✓ | |
| **Precise IF Judge** | | ✓ | | | |
| **Math/Code Judge** | | | ✓ | | |
| **Factuality Judge** | ✓ | | | | |
| **Safety Judge** | | | | | ✓ |

Table 1: Tasks and their corresponding RM and MoJs

**Baseline and Ablations:** We conducted CGPO training with all three optimizers that we proposed: CRPG, CRRAFT, and CODPO. Additionally, we consider DPO and PPO as our RLHF baselines. To establish the DPO baseline, we continue running the DPO updates starting from the RLHF warm-up model and extend the training steps to thoroughly optimize all evaluation benchmarks. To establish the PPO baseline, we first train a unified reward model by merging all reward models' training data. Following this, we start from the RLHF warm-up model and perform PPO updates by applying the unified reward model to the same prompt sets as CGPO recipes.

## 4.2 Main Results and Ablations

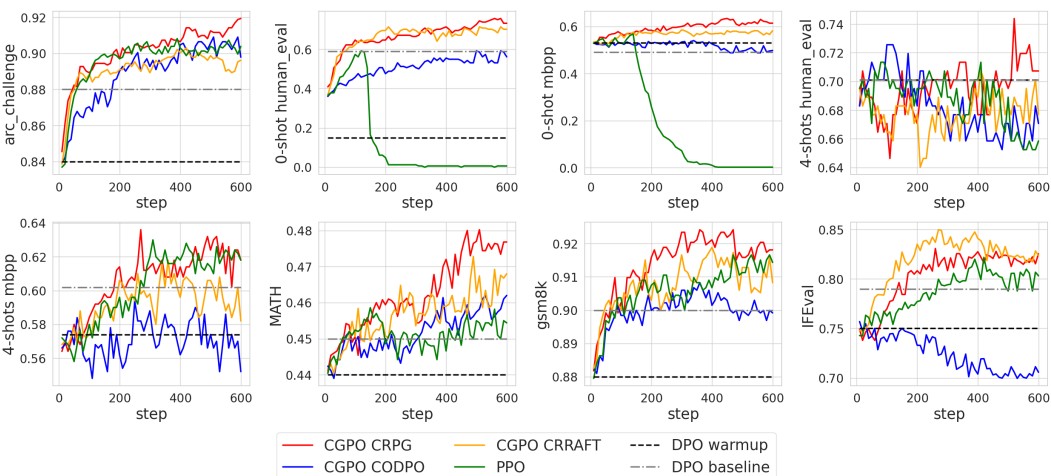

Figure 3: Comparison of CGPO variants with baseline RLHF algorithms PPO and DPO

For the online RLHF algorithms CGPO and PPO, we monitor the model's performance at every 10-step interval throughout the training trajectory across various benchmarks, as illustrated in Figure 3. The plot demonstrates that CGPO, when paired with the CRPG and CRRAFT optimizers, consistently enhances performance across all benchmarks compared to the initial model, indicating progressive improvement as training progresses. Specifically, CRPG outperforms all others throughout the entire training period in terms of ARC Challenge, 0-shot HumanEval, 0-shot MBPP, 4-shots MBPP, MATH, and GSM8K. Meanwhile, CRRAFT excels in IFEval during the training phase. Notably, the online RLHF baseline PPO exhibits a significant decline in performance on 0-shot coding benchmarks (MBPP and HumanEval) as training progresses, indicating a severe case of reward hacking. Meanwhile, CGPO with the CODPO optimizer shows a slight regression on MBPP and IFEval benchmarks compared to the warm-up model, yet it effectively avoids the drastic performance drop observed with PPO in the coding benchmarks. The offline RLHF baseline DPO, while avoiding the drastic regression seen with PPO, remains overly conservative in enhancing the model's performance, resulting in lower metric improvements compared to CGPO with the CRPG and CRRAFT optimizers.

In Table 2, we present the evaluation results for SFT, DPO warm-up, DPO baseline, the final step of PPO, and various CGPO variants across all benchmarks detailed. The data in Table 2 indicate that CGPO variants employing CRPG and CRRAFT optimizers significantly outperform the DPO and PPO baselines across all benchmarks. Notably, CRPG shows the most substantial improvements in math and coding benchmarks (Math, GSM8K, HumanEval, and MBPP), while CRRAFT excels in helpfulness and factuality (AlpacaEval-2, Arena-Hard, and TruthfulQA). Both CRPG and CRRAFT achieve the best results in terms of instruction following (IFEval). While the CGPO variant with the CODPO optimizer does not perform as strongly as other variants, it offers performance that is on par with or better than the DPO and PPO in all benchmarks except the IFEval. In terms of safety, CGPO with the CRPG and CODPO optimizers achieve the best results in FRR and SVR, respectively. Table 2 demonstrates that the CGPO framework is able to enhance model quality across all tasks, proving its efficacy in managing challenging multi-task fine-tuning.

### 4.3 Effectiveness of Mixture of Judges

In this section, we explore the significance of incorporating MoJs within the CGPO framework. We conduct an ablation study by eliminating all MoJs from CGPO, utilizing the CRPG optimizer, while keeping all other variables constant, and then proceed to rerun the RLHF finetuning for 600 steps. Figure 4 presents a comparative analysis of CGPO performance with and without MoJs using the CRPG optimizer across various benchmarks, including HumanEval, MBPP, MATH, and GSM8K.

|  | DPO warm-up | DPO baseline | PPO | CGPO - CRPG | CGPO - CRRAFT | CGPO - CODPO |
|---|---|---|---|---|---|---|
| AlpacaEval-2 | 13.3 | 16.3 | 24.8 | 25.9 | **43.2** | 18.08 |
| Arena-Hard | 18.8 ± 1.6 | 18.3 ± 1.7 | 24.3 ± 1.8 | 31.2 ± 2.2 | **36.8 ± 2.0** | 16.8 ± 1.9 |
| IFEval | 0.75 | 0.79 | 0.81 | **0.83** | **0.83** | 0.70 |
| MATH | 0.44 | 0.45 | 0.46 | **0.48** | 0.47 | 0.46 |
| GSM8K | 0.88 | 0.90 | 0.91 | **0.93** | 0.92 | 0.90 |
| 0-shot MBPP | 0.51 | 0.49 | 0.002 | **0.63** | 0.57 | 0.51 |
| 4-shots MBPP | 0.57 | 0.60 | **0.62** | **0.62** | 0.58 | 0.55 |
| 0-shot HumanEval | 0.15 | 0.59 | 0.006 | **0.76** | 0.70 | 0.57 |
| 4-shots HumanEval | 0.70 | 0.70 | 0.66 | **0.71** | 0.68 | 0.67 |
| MMLU | **0.76** | 0.75 | 0.75 | 0.75 | 0.75 | 0.75 |
| ARC | 0.84 | 0.88 | 0.90 | **0.92** | 0.90 | 0.90 |
| TruthfulQA | 0.59 | 0.63 | 0.65 | 0.64 | **0.66** | 0.63 |
| Engagement | 0.59 | 0.71 | **0.81** | **0.81** | 0.72 | 0.79 |
| SVR | 0.03 | 0.02 | 0.03 | 0.05 | 0.02 | **0.01** |
| FRR | 0.161 | 0.17 | 0.12 | **0.04** | 0.12 | 0.24 |

Table 2: Evaluation results of SFT, DPO warm-up, DPO, PPO and CGPO variants

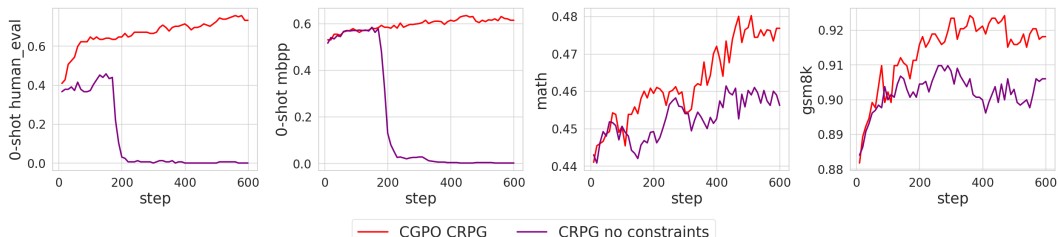

Figure 4: Comparison of CGPO (CRPG optimizer) with and without MoJs

From Figure 4, it is clear that in the absence of coding judges, the CRPG optimizer undergoes a notable decline in 0-shot coding benchmarks once it surpasses 180 steps, mirroring the performance of the PPO baseline. Additionally, in the MATH and GSM8K, while CRPG shows some improvement without constraints, the increases in metrics are considerably less pronounced compared to cases where math judges are utilized. This comparison effectively illustrates that MoJs play a crucial role not only in preventing reward hacking but also in significantly boosting the model's performance during online RLHF finetuning.

## 5 Conclusion

In this paper, we introduced the CGPO framework to address key challenges in multi-task learning for LLM post-training with RLHF. The CGPO framework effectively mitigates issues such as inhomogeneous reward hacking and conflicting task goals through a novel primal-type multi-constraint RL method and a tailored multi-objective optimization strategy. We demonstrate the effectiveness of CGPO in a scenario where we need to handle five tasks with three reward models and six constraints, marking the first application of RLHF in multi-task learning for general-purpose LLMs. Our experiments show that CGPO achieves significantly better metric gains for all tasks compared to the baseline RLHF methods. Moving forward, it is promising to explore more automated ways to adapt the gradient weights from different tasks to further reduce the hyperparameter tuning burden and advance the Pareto frontier (Sener & Koltun, 2018).

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
