## A  RELATED WORKS

**RLHF in MTL.** Reinforcement Learning with Human Feedback (RLHF) is designed to align language models with human preferences and has become a crucial component of the fine-tuning pipeline for Large Language Models (LLMs) (Stiennon et al., 2020; Ouyang et al., 2022; Brown et al., 2020; Touvron et al., 2023; Bi et al., 2024; Bai et al., 2022). The majority work of RLHF focus optimizing a single reward models (Ouyang et al., 2022; Gao et al., 2023; Dong et al., 2023; Ethayarajh et al., 2023). The exploration of RLHF in the MTL setting remains relatively under-explored. The most commonly adopted approach involves optimizing a weighted sum of several reward models, where each model captures the interests of different tasks (Ramamurthy et al., 2022; Glaese et al., 2022; Yuan et al., 2023; Bakker et al., 2022; Wu et al., 2024). However, a major limitation of this approach is that key information from each individual reward model can be lost through linear combination, particularly when conflicting task goals exist. This can lead to suboptimal performance for each individual task. Additionally, each individual reward model typically requires different treatments (regularization, early stopping, etc) due to their unique properties, thus applying a uniform treatment for a composite reward model can further impair optimization performance across tasks (Moskovitz et al., 2023). Another research direction involves fine-tuning a separate LLM model for each task, followed by linear interpolation of the LLM weights across all learned models to produce a single model that excels in multiple tasks (Rame et al., 2024). However, this method remains computationally expensive and unstable due to the high cost and variability inherent in a single RLHF process (Hu et al., 2023; Rafailov et al., 2024b). (Yang et al., 2024) Proposed to use in-context reward model to manage multiple reward, but introduce additonal cost during inference time. Unlike the approaches mentioned above, CGPO introduces a customized reward model recipe and an RLHF optimizer tailored for each specific task. This method is not only as efficient as the conventional RLHF pipeline, but it also preserves all information within each reward model, thereby optimizing alignment for each task to the fullest extent.

**Reward Hacking Mitigation.** Compaired with traditional RL, where the reward is typically well-defined and the goal is to maximize it (Sutton & Barto, 2018), RLHF introduces a unique challenge known as "reward hacking." This issue arises because the reward model serves as a proxy for actual human preferences. Over-optimization of the reward model can adversely impact the performance of the language model (Gao et al., 2023; Moskovitz et al., 2023; Stiennon et al., 2020; Rafailov et al., 2024b). Consequently, addressing reward hacking is a major focus in RLHF. Previous studies have explored various approaches to mitigate the effects of reward hacking, including reward model regularization (Singhal et al., 2023), reward ensembles (Eisenstein et al., 2023; Ramé et al., 2024), and explicitly learning the reward bias error (Chen et al., 2024; Shen et al., 2023). In contrast to previous methods, our CGPO framework employs both LLM and rule-based judges as constraints to detect and prevent reward hacking patterns. This approach offers a more fine-grained and controllable solution to this persistent issue. Furthermore, the use of MoJs enables us to develop tailored strategies for mitigating the effects of reward hacking across various tasks in the MTL setting. This allows us to effectively address the reward hacking challenge in the more complex MTL environment, where previous methods have struggled to perform efficiently.

## B  CGPO OPTIMIZERS

### B.1  CALIBRATED REGULARIZED POLICY GRADIENT (CRPG)

In this section, we discuss our new constraint RLHF optimizer, the Calibrated Regularized Policy Gradient (CRPG), which is a policy gradient-based approach.

**Calibrated Reward.** In the traditional RLHF algorithm, the reward model is typically directly incorporated into RL optimizers to progressively refine the policy. However, this method can pose difficulties when the reward model value is not properly calibrated. For preference reward models trained with eq. (1), the reward's accuracy may be proficient in distinguishing between good and bad generations from the same prompt. However, the reward model values between generations from different prompts may not be directly comparable due to potential significant variations in the reward model value range for different prompts. Due to such reasons, standard RLHF algorithms, such as PPO and REINFORCE, could lead to suboptimal performance due to the poor calibration of

the reward model (Rita et al., 2024). In CRPG, we introduce a novel and low-cost reward calibration strategy to address this issue.

We consider the scenario where each prompt $s$ used in RLHF fine-tuning has a corresponding baseline response $\bar{a}$. This condition can be easily satisfied in practice.

- **Option 1:** We repurpose the prompt set from the SFT training set and/or the reward model training set. For the SFT training dataset, the pre-collected golden response is utilized as the baseline response, denoted as $\bar{a}$. For the pair-wise reward model training dataset, the preferred response is designated as the golden response $\bar{a}$.

- **Option 2:** Given an RLHF fine-tuning prompt set $D_d$, we use $\pi_{\text{ref}}$ to generate the baseline response for all prompts $s \in D_d$, i.e., $\bar{a} \sim \pi_{\text{ref}}(\cdot|s)$ before starting RLHF fine-tuning.

Without loss of generality, we assume there is an underlying policy $\bar{\pi}$ that generates the baseline responses, denoted as $\bar{a} \sim \bar{\pi}(\cdot|s)$. Given the baseline response $\bar{a}$, we developed the following calibrated reward to replace the raw reward model $r_\phi(s, a)$:

$$R_{calib}(s, a) = \sigma(r_\phi(s, a) - r_\phi(s, \bar{a})). \tag{4}$$

Intuitively, $R_{pair}(s, a)$ here represent the probability of $a$ being better than baseline response $\bar{a}$ conditioned on the same prompt $s$, i.e.,

$$R_{calib}(s, a) \approx \text{Prob}(a > \bar{a}|s).$$

The advantages of using calibrated rewards $R_{calib}$ are twofold:

1. The magnitude of $R_{calib}$ becomes meaningfully comparable across different prompts. This is because it represents the probability that the current policy $\pi$ is superior to the baseline $\bar{\pi}$ for different actions. In other words, if $R_{calib}(s, a) > R_{calib}(s', a')$, it directly implies that action $a$ given state $s$ is better than action $a'$ given state $s'$, conditioned on the baseline policy $\bar{\pi}$. However, this implication cannot be made if $r_\phi(s, a) > r_\phi(s', a')$.

2. The magnitude of the calibrated reward model is strictly bounded between 0 and 1. This constraint prevents an action with an extremely large raw value from dominating the policy update direction, which could be misleading, since a large raw reward value does not necessarily imply superior action quality.

Based on $R_{calib}(s, a)$, we now reformulate RLHF objective in eq. (2) as

$$\max_w \bar{J}(\pi_w) = \mathbb{E}_{a \sim \pi_w(\cdot|s), s \sim D_d} [R_{calib}(s, a)] \tag{5}$$

where $\bar{J}(\pi_w)$ is the policy optimization objective. Intuitively, it represents the probability of current policy $\pi_w$ being better than the baseline policy $\bar{\pi}$ conditioned on the prompt set $D_d$, i.e.,

$$\bar{J}(\pi_w) \approx \text{Prob}(\pi_w > \bar{\pi}|D_d).$$

**Constraint Regularized Gradient.** Recall that in the multi-constraint setting, our goal is to maximize the expected reward model while aligning the LLM such that its generations strictly adhere to a set of constraints. These constraints compensate for the limitations of the reward model, including safety requirements, reasoning accuracy, and factual correctness. These aspects may not be fully captured by the reward model but can be well addressed via a separate rule-based judge or an LLM-based judge. Note that the "Positive samples" in line 6 of Algorithm 1 is a subset of $\Sigma$, i.e., $X_t^+ \in \Sigma$. Consequently, we aim to optimize the following multi-constraint objective, denoted as $\bar{J}_c$:

$$\max_w \bar{J}_c = \mathbb{E}_{a \sim \pi_w(\cdot|s), s \sim D_d} [R_{calib}(s, a) \cdot \mathbf{1}_{(s,a) \in \Sigma}]. \tag{6}$$

By solving the optimization problem presented in eq. (6), the LLM is aligned to maximize the expected value of the calibrated reward model as much as possible, while remaining within the constraint satisfaction region.

Given $R_{calib}$ and $\Sigma$, we define the following constraint regularized reward as

$$R_{cr}(s, a) = \begin{cases} R_{calib}, & \text{if } (s, a) \in \Sigma \\ 0, & \text{if } (s, a) \notin \Sigma \end{cases} \tag{7}$$

With the calibrated regularized reward $R_{cr}$, we rewrite eq. (6) as

$$\max_w \bar{J}_c = \mathbb{E}_{a \sim \pi_w(\cdot|s), s \sim D_d} \left[ \cdot R_{cr}(s, a) \right]. \tag{8}$$

We consider the following update to optimize $\bar{J}_c$

$$w_{t+1} = w_t + \alpha_t \cdot g_c(\pi_{w_t}), \tag{9}$$

where

$$g_c(\pi_w) = \frac{1}{N} \sum_i^N \nabla \log \pi_w(s_i, a_i) \cdot R_{cr}(s_i, a_i).$$

**CRPG Implementation.** Consider the KL divergence between $\pi_{\text{ref}}$ and $\pi_w$ as a universal regularization method to prevent reward hacking during CRPG fine-tuning. We propose the following new reward regularization approach:

$$\tilde{R}_{cr}(s, a) = \max \left\{ 1 - \frac{\log(\pi_w(s_i, a_i)/\pi_{\text{ref}}(s_i, a_i))}{\text{KL}_{\max}}, 0 \right\} \cdot R_{cr}(s, a). \tag{10}$$

It is important to note that $\tilde{R}_{cr}$ not only penalizes samples that deviate significantly from $\pi_{\text{ref}}$, but also strictly bounds the overall KL divergence.

Moreover, to reduce the variance in the CGPG gradient estimation, we consider subtracting a baseline from the $g_c$ without changing its expected direction as following

$$\tilde{g}_c(\pi_{w_t}) = \frac{1}{n} \sum_i^n \nabla \log \pi_{w_t}(s_{t,i}, a_{t,i}) \cdot \left[ \tilde{R}_{cr}(s_{t,i}, a_{t,i}) - \frac{1}{n} \sum_i^n \tilde{R}_{cr}(s_{t,i}, a_{t,i}) \right]. \tag{11}$$

The final CRPG update in multi-constraints finetuning setting is given as

$$w_{t+1} = w_t + \alpha_t \cdot \tilde{g}_c(\pi_{w_t}). \tag{12}$$

## B.2 Constraint Regularized Reward Ranking Finetuning (CRRAFT)

In this section, we introduce another constrained RLHF policy optimizers that we proposed: Constraint Regularized Reward Ranking Finetuning (CRRAFT), which is built upon the RAFT.

In the original RAFT algorithm (Dong et al., 2023), each round involves generating multiple responses from a prompt using the current policy model, denoted as $\{a_{t,i}^1, a_{t,i}^2, \ldots, a_{t,i}^K\} \sim \pi_{w_t}(\cdot \mid s_{t,i})$. A reward model $r$ is then utilized to select the response with the highest reward model score, i.e., $a_j^* = \text{argmax}_{k \in [K]} r_{\text{pair}}(s_{t,i}, a_{t,i}^k)$ (note that whether a calibrated reward is used or not does not affect the reward ranking result). Subsequently, an one-step SFT update is performed to maximize the likelihood of this generated sample $(s_{t,i}, a_{t,i}^*)$. The policy model is iteratively updated to improve its alignment with the reward model $r_{pair}$ as follow

$$w_{t+1} = w_t + \alpha_t \cdot \frac{1}{n} \sum_{j=1}^n \nabla \log(\pi_{w_t}(s_{t,i}, a_{t,i}^*)). \tag{12}$$

In the multi-constraint setting, we make the following two changes on top of RAFT to develop our CRRAFT optimizer:

- After applying the reward model to score each responses, we adopt Option I in Algorithm 1 to first filter out those generated responses that violated any of the constraints. Additionally, to avoid large drift of current policy from starting point policy $\pi_{\text{ref}}$, we also filter out all generations whoes KL-divergence is larger than a pre-defined threshold $\text{KL}_{\max}$, i.e., $\text{KL}_{(s_{i,t}, a_{i,t}^k)} = \frac{\log \pi_{w_t}}{\log \pi_{\text{ref}}}(s_{i,t}, a_{i,t}^k) > \text{KL}_{\max}$. After that we apply reward ranking to select the one with the highest reward model score from the rest of responses, i.e.,

$$a_{i,t}^* = \underset{\substack{k \in [K], \\ (s_{i,t}, a_{i,t}^k) \in X_t^+, \\ \text{KL}_{(s_{i,t}, a_{i,t}^k)} \leq \text{KL}_{\max}}}{\text{argmax}} r_\phi(s_{i,t}, a_{i,t}^k). \tag{13}$$

We refer to the procedure in eq. (13) as constrained regularized reward ranking. It's important to note that CRRAFT not only has the capability to manage multiple constraints, but it also strictly bounds the KL-divergence. This is a feature that the standard RAFT algorithm lacks.

Note that there may be instances where no generations remain after filtering. In such cases, if the pre-collected baseline response $\bar{a}_{i,t}$ satisfies all constraints, it can be used as $a_{i,t}^*$. If it doesn't, this datapoint can be skipped.

- After the constrained regularized reward ranking, instead of directly performing SFT update w.r.t the chosen sample as eq. (12) does, here we reweigh each chosen response by their calibrated reward value and then perform SFT update as follow

$$w_{t+1} = w_t + \alpha_t \cdot \tilde{g}_{ra}(\pi_{w_t})$$

$$= w_t + \alpha_t \cdot \frac{1}{n} \sum_{i=1}^{n} R_{calib}(s_{i,t}, a_{i,t}^*) \cdot \nabla \log(\pi_{w_t}(s_{i,t}, a_{i,t}^*)). \tag{14}$$

By incorporating the calibrated reward model value in the update, we can differentiate the emphasis on chosen responses based on their quality, unlike the RAFT algorithm which treats all chosen responses equivalently. This approach allows for a more refined alignment with the reward model.

Please note that unlike CRPG, CRRAFT specifically focuses on increasing the likelihood of constraint-satisfied positive samples and disregards the constraint-violated negative samples.

### B.3 Constrained Online Direct Preference Optimization (CODPO)

Based on Direct Preference Optimization (DPO), a widely used offline RLHF alignment algorithm in the unconstrained setting, we propose a new variant called Constrained Online Direct Preference Optimization (CODPO) to solve the constrained RLHF fine-tuning problem.

Recall that in DPO (Rafailov et al., 2024b), the optimal policy $\pi^*$, which aligns with human preferences in the $\beta$-regularized MDP setting, satisfies the following preference model:

$$P_{\pi^*}(a_p > a_n) = \frac{1}{1 + \exp\left(\beta \log \frac{\pi^*(s, a_n)}{\pi_{\text{ref}}(s, a_n)} - \beta \log \frac{\pi^*(s, a_p)}{\pi_{\text{ref}}(s, a_p)}\right)}.$$

Given a pairwise preference sample pair $(s, a_p)$ and $(s, a_n)$, we update our policy by solving the following problem:

$$\min_w \mathcal{L}_{\text{DPO}}(\pi_w) = -\mathbb{E}_{(s, a_p, a_n)}\left[\ell_{\text{DPO}}(\pi_w, s, a_p, a_n)\right].$$

$$\text{where} \quad \ell_{\text{DPO}}(\pi_w, s, a_p, a_n) = \log \sigma\left(\beta \log \frac{\pi_w(s, a_p)}{\pi_{\text{ref}}(s, a_p)} - \beta \log \frac{\pi_w(s, a_n)}{\pi_{\text{ref}}(s, a_n)}\right) \tag{15}$$

To prevent the possible decreasing likelihood of positive samples $a_p$, it has been proposed to add a regularization term to the vanilla DPO loss (Pal et al., 2024):

$$\tilde{\ell}_{\text{DPO}}(\pi_w, s, a_p, a_n) = \ell_{\text{DPO}}(\pi_w, s, a_p, a_n) + \frac{\lambda}{|a_p|} \cdot \log(\pi_w(s, a_p)), \tag{16}$$

where $|a_p|$ represents the length of response $a_p$. By appropriately tuning the hyperparameter $\lambda$, the formulation in eq. (16) can effectively increase the likelihood of $a_p$ while decreasing the likelihood of $a_n$ to maximize the margin between positive and negative generations.

In CODPO, similar to CRRAFT, we first generate multiple responses for each prompt using the current policy $\{a_{t,i}^1, a_{t,i}^2, \dots, a_{t,i}^K\} \sim \pi_{w_t}(\cdot \mid s_{t,i})$ and split the generations into positive samples $X_t^+$ and negative samples $X_t^-$. After that, we select the positive sample from $X_t^+$ with the highest reward value, and the negative sample from $X_t^-$ with the lowest reward value, i.e.,

$$a_{i,t}^+ = \operatorname*{argmax}_{\substack{k \in [K], \\ (s_{i,t}, a_{i,t}^k) \in X_t^+}} r_\phi(s_{i,t}, a_{i,t}^k),$$

$$a_{i,t}^- = \underset{\substack{k \in [K], \\ (s_{i,t}, a_{i,t}^k) \in X_i^-}}{\text{argmin}} \ r_\phi(s_{i,t}, a_{i,t}^k).$$

In cases where no generations satisfy all constraints, we can skip this sample. Conversely, when no generations violate any constraints, we can select the generation with the lowest reward model value as the negative sample.

Then, at each iteration, we update the policy as follows:

$$w_{t+1} = w_t - \alpha_t \cdot \frac{1}{n} \sum_{i=1}^{n} \nabla \tilde{\ell}_{\text{DPO}}(\pi_{w_t}, s_{i,t}, a_{i,t}^+, a_{i,t}^-). \tag{17}$$

## C  EXPERIMENT DETAILS

### C.1  MULTI-TASKS LEARNING

In this work, we focus on fine-tuning a LLM to achieve alignment across the following five tasks:

- **General chat:** This task is designed to enhance the general conversational abilities of LLMs by considering multi-turn conversational histories (Wang et al., 2024). It focuses on boosting the coherence, consistency, and correctness of responses, thereby making the interactions more logical and seamless. Additionally, this task improves the model's capability to deliver responses that are better aligned with the user's intentions and queries, and are factually grounded (Sun et al., 2024).

- **Instruction Following:** This task is designed to enhance the ability of LLMs to follow instructions accurately within specific contexts or industries (Zhou et al., 2023). By fine-tuning LLMs to adapt to particular domains or user requirements, they can deliver more precise and relevant responses. This improvement leads to a more satisfying and efficient user experience, making LLMs more effective and versatile tools across various applications.

- **Math/Code Reasoning:** This task is designed to enhance the math and coding capabilities of LLMs, enabling them to address more complex problems and broaden their range of functions. These include tasks like debugging code or solving mathematical equations, which are vital in technical fields (Hendrycks et al., 2021b; Cobbe et al., 2021; Chen et al., 2021; Austin et al., 2021). Furthermore, improving LLMs' ability to comprehend and produce mathematical and code-related content results in greater accuracy and efficiency in activities that demand meticulous logical reasoning and computational thinking.

- **Engagement Intent:** This task aims to enhance user engagement and interaction with the LLM. To address this, we involve human annotators who interact with the model and provide binary feedback (like or dislike) for each response generated by the LLM. Our objective is to maximize the likelihood that users will favorably respond to the LLM's outputs.

- **Harmful Intent:** This task trains LLMs to recognize and resist safety-related adversarial attacks. It ensures that LLMs are safeguarded against exploitation for malicious purposes, such as generating harmful or misleading information (Sun et al., 2024; Xu et al., 2020). By enhancing their ability to operate safely and ethically, this task helps maintain user trust and uphold the credibility of the technology.

### C.2  SUPERVISED FINE-TUNING

The foundational model we have chosen is the LLaMA-3.0-70B pre-trained checkpoint. We independently perform SFT using an open-source dataset to establish the initial policy, denoted as $\pi_0$. For all preference pair datasets listed below we only use positive samples in SFT. We utilize the following datasets for the tasks under consideration:

- **General chat:** LMSys-55k (Chiang et al., 2024), UltraChat (Ding et al., 2023)
- **Instruction following:** LIama 3.0 70B instruct model synthetic instruction following dataset

- **Math/Code Reasoning:** Orca-Math Mitra et al. (2024), MetaMath (Yu et al., 2023), Evol-CodeAlpaca (Luo et al., 2023), UltraFeedback (Cui et al., 2023), UltraInteract (Yuan et al., 2024a)
- **Harmful Intent:** Human annotated safety dataset

The training is carry out for 2 epoches with a learning rate of $10^{-5}$. A cosince schedule is employed, the global batchsize is set to 128 with minimum rate 0.1 and warm-up steps 200. The detail of how we obtain synthetic instruction following dataset and safety dataset SFT can be found in Appendix C.6.

## C.3 REWARD MODELLING

We have employed open-source pairwise preference data to train three specialized reward models (RMs):

- **Helpfulness RM:** This model is tailored for tasks such as general chat, instruction following, and math/code reasoning. It is based on the LLaMA-3-70B instruct finetuned model. The training utilized the following pairwise preference datasets:
  - **General chat:** Includes datasets such as HH-RLHF (Bai et al., 2022), SHP (Ethayarajh et al., 2022), HelpSteer (Wang et al., 2023), Distilabel-Capybara (Ethayarajh et al., 2024), Distilabel-Orca (Álvaro Bartolomé Del Canto et al., 2024), and LMSys-55k (Chiang et al., 2024).
  - **Instruction Following:** LIama 3.0 70B instruct model synthetic instruction following pairwise preference dataset.
  - **Math/Code Reasoning:** Features datasets like Argilla Math (Álvaro Bartolomé Del Canto et al., 2024), UltraFeedback (Cui et al., 2023) and UltraInteract (Yuan et al., 2024a).
- **Engagement RM:** This RM is designed to simulate user engagement preferences. Initially, we fine-tune a binary classifier predictor using the LLaMA-3-70B instruct model to predict a user's engagement intent based on real interaction data between the language model and the user. We then treat this predictor as the oracle for user intent regarding engagement with the language model, given prompts and generations. To gather pair-wise training data, we subsample 129692 prompts from the LMSys-1M dataset (Zheng et al., 2023a) and use the LLaMA-3-70B instruct model to generate four responses for each prompt. Each prompt is then scored using the oracle engagement predictor. We select the generation with the highest score as the "chosen" response and the generation with the lowest score as the "rejected" response. By doing this, we compile the pair-wise dataset and train the engagement RM based on this data.
- **Safety RM:** Focused on ensuring safe responses in scenarios with potentially harmful user prompts, this model is based on the LLaMA-3-8B instruct finetuned model. It utilizes a human-annotated safety pairwise preference dataset that identifies harmful intent in prompts.

It is important to note that we are considering training a unified Helpfulness RM that encompasses general chat, instruction following, and math/code reasoning, rather than training three separate RMs. This consideration is based on the observed positive correlation among these tasks. A unified RM, trained with a blended dataset from these domains, is expected to yield superior performance compared to training separate RMs for each individual task.

## C.4 MIXTURE OF JUDGES

To address the limitations of the reward model, we have implemented several judges in our experiment for multi-task alignment:

- **Precise instruction following judge:** Reward models often struggle with precisely following instructions (Zhou et al., 2023). To address this, we have implemented a rule-based judge capable of accurately assessing compliance with over 30 types of specific instruction-following requests found in user prompts, such as "answer the question in two paragraphs."

It is important to note that during RLHF finetuning, we will also include precise instruction-following prompts of this type so that the correctness of the generation can be evaluated with this constraint judge.

- **Regex math/code reasoning judge:** Reward models frequently fail to accurately assess the correctness of math and coding problems. To improve accuracy, we have introduced specialized judges for both domains. For math-related queries, we use a rule-based approach to check whether the final answers of responses match the ground-truth answers. For coding problems, we employ a unit-test-based judge that evaluates the accuracy of the code by running it through a series of unit tests.

- **False refusal judge:** Enhancing safety protocols may cause LLMs to become overly safe, leading to false refusals when responding to innocuous user queries, thus degrading user experience. It has become critical for LLMs to reduce false refusals while maintaining the same level of safety, both in the research community and in the leading industry models (Cui et al., 2024). To address this challenge, we have developed a false refusal classifier, a fine-tuned LLM designed to detect false refusals to ensure the effectiveness of the LLM.

- **Factuality judge:** Hallucination is a common issue in LLMs, especially during the RLHF phase. The reward model often fails to distinguish between factual and non-factual claims. To address this, we use the Llama3 70B model as a factuality constraint judge to evaluate whether the fact-related claims in an output contradict pre-collected, verified factual data, thereby ensuring the accuracy and reliability of the information provided by the LLM.

- **Safety judge:** The safety reward model alone does not sufficiently ensure the trustworthiness of our model due to its limited accuracy. To further enhance safety, we incorporate LlamaGuard2, an industry leading open sourced fine-tuned LLM, to assess whether an output violates predefined safety standards.

In this section, we will next discuss in detail about how we build MoJs in CGPO in our experiment.

### C.4.1 RULE-BASED CONSTRAINT JUDGE

**Precise Instruction following judge.** The precise instruction-following constraint judge begins by reading the metadata to understand the specific rules that LLM's output must adhere to. Then, we employ string-matching based logic to determine whether LLM's generation complies with all the specified rules.

**Math judge.** Similar to the instruction-following judge, our math judge also employs string-matching logic to verify the correctness of the LLM's response by comparing it with the ground truth answer provided in the metadata.

**Coding judge.** Our coding constraint judge examines the coding block in LLM's response to extract the code snippet. It then runs the snippet through all the unit tests provided in the metadata to determine if it passes each test. Similar to the math constraint, false negatives can occur if LLM's solution is not formatted correctly. Implementing CGPO to discourage such patterns could enhance the model's ability to follow instructions accurately.

### C.4.2 LLM-BASED CONSTRAINT JUDGE

The LLM classifier constraint judge utilizes an additional LLM to assess whether the output from our training LLM adheres to a specific predefined criterion. We design the input for this judge using a prompt template that arranges the LLM's response alongside other essential contexts. Within this template, we specify both a negative token and a positive token. The negative token indicates that the LLM's response breaches the constraint, while the positive token signifies compliance. We explicitly direct the judge to issue either the positive or negative token based on their assessment. To minimize the randomness in the judgment process, we do not rely solely on the LLM to generate a token and then check its correspondence to the negative or positive token. Instead, we directly examine the softmax probabilities of the negative and positive tokens. If the probability of the negative token is higher, we conclude that the LLM's response violates the constraint, and vice versa. Table 4 presents the template along with the negative and positive tokens for the LLM classifiers in our experiment.

**False refusal constraint judge.** We utilize the Llama 3.0 8b pretrained model as a foundation and fine-tune an LLM classifier specifically aimed at identifying refusal patterns in LLM responses.

The training data is formatted as follows: "[INST] {LLM response} [\INST] judgment", where "judgment" is True if the LLM response indicates refusal, and False otherwise. During the inference phase of deploying this constraint judge, we also encapsulate the generated responses from the training LLM within "[INST] ... [\INST]" and use that as the input for the judge.

**Factuality constraint judge.** We employ the Llama 3.0 70b instruct model directly as the factuality constraint judge. Recall that for prompts associated with deterministic factuality, we include the ground truth answer in the metadata. When deploying this constraint judge, we use the template as illustrated in Table 4, incorporating the prompt, ground truth answer, and the LLM response into the template to serve as inputs for the judge.

**Safety constraint judge.** We utilize LIamaGuard2, which is fine-tuned from the Llama 3.0 8b pretrained model. We reuse the template as introduced in the LIamaGuard2 paper, where we incorporate pre-defined safety guidelines and full completions into the prompt template to serve as inputs for the judge.

C.5    CGPO PROMPT SETS

We list the prompt set that we used for each tasks in our experiments as following:

- **General Chat:** UltraChat, LMSys-55k, XSTest, TriviaQA, ARC
- **Instruction Following:** Synthetic IF prompts
- **Math/Coding Reasoning:** Math, GSM8K, Aqua, APPS
- **Engagement Intent:** LMSys-1M
- **Harmful Intent:** Safety RM training prompt

C.6    DETAIL OF TRAINING DATASETS

The detail of of our training dataset is provide in Table 3. Note that in our experiment we adopt the instruction finetuing format, in which the prompt is wrapped as "[INST] {prompt} [\INST]":

**Synthetic IF dataset.** Inspired by Zhou et al. (2023), we consider synthetic prompts that require LLM generation to satisfy one or more closed-form instructions, which can be verified exactly. We identify 23 types of closed-form instructions for generation and use LIama 3.0 70B instruct model to create synthetic prompts that address a specific topic and also require these closed-form instructions. We create a template to enable LIama 3.0 70B instruct model to generate all prompts. The prompt template that we input into LIama 3.0 70B instruct model to generate synthetic instruction-following prompts is provided as follows:

```
Prompt Template =

"You are a helpful AI assistant.  You are given a TOPIC and a FORMAT
REQUIREMENT, and you are expected to generate a PROMPT that is on
the given TOPIC and specify the given FORMAT REQUIREMENT that the
corresponding answer should follow.  Here are many examples that you
can learn from:

TOPIC: Travel

FORMAT REQUIREMENT: In your entire response, refrain from the use of any
commas

PROMPT: I am planning a trip to Japan, and I would like thee to write an
itinerary for my journey in a Shakespearean style.  You are not allowed
to use any commas in your response.

TOPIC: Aerospace engineering
```

**FORMAT REQUIREMENT:** In your entire response, refrain from the use of any commas and Give two different responses. Responses and only responses should be separated by 6 asterisk symbols: ******

**PROMPT:** Write two jokes about rockets. Do not contain commas in your response. Separate the two jokes with 6 asterisk symbols: ******.

**TOPIC:** History

**FORMAT REQUIREMENT:** Entire output should be wrapped in JSON format

**PROMPT:** What is the history of NYC prospect park? Please wrap your entire answer in JSON format.

**TOPIC:** Video game

**FORMAT REQUIREMENT:** Highlight at least 2 sections in your answer with markdown, i.e. *highlighted section* and Answer with at least 40 sentences

**PROMPT:** Can you write a poem about the pros and cons of playing a lot of video games? Please make sure it's at least 40 sentences long (don't forget to add punctuations). You must highlight at least sections in your response, like *highlighted phrase*.

**TOPIC:** Movie

**FORMAT REQUIREMENT:** Answer with at least 40 sentences, Highlight at least 4 sections in your answer with markdown, i.e. *highlighted section*, and Wrap your entire response with double quotation marks

**PROMPT:** Write a joke about the superhero movie with at least 5 sentences. Use Markdown to italicize at least 4 sections in your answer, i.e. *italic text*. Wrap your answer in double quotes.

**TOPIC:** Health care

**FORMAT REQUIREMENT:** Your entire response should be in English, capital letters only

**PROMPT:** Write an essay about public health care system in US in English and in all capital letters.

**TOPIC:** Mathematics

**FORMAT REQUIREMENT:** Entire output should be wrapped in JSON format

**PROMPT:** List all facts about calculus in a structured output. In particular, Format your entire output in JSON.

Now it is your turn to generate a PROMPT that is on the given TOPIC and specify the given FORMAT REQUIREMENT that the corresponding answer should follow. Please DO NOT make up any new format requirement that is not given to you.

**TOPIC:** {topic}

**FORMAT REQUIREMENT:** {instruction}

```
To be noted, you just need to mention/specify the FORMAT REQUIREMENT
in your response but your response does not need to follow it.  Please
directly provide the PROMPT without any extra words.  Do not write any
note or explanation.

TOPICS = ["20th century events", "Accounting", "Architecture",
"Astronomy", "Biology", "Businessethics","Celebrities","Chemistry","Clinical
knowledge", "Economics", "Electrical engineering", "Ethics of artificial
intelligence", "Education", "Energy", "Gaming", "Geography", "Global
facts", "History", "Healthcare", "Immigration law", "International
law", "Jurisprudence", "Management", "Marketing", "Mathematics",
"Medicine", "Moraldisputes", "Movies", "Music", "Philosophy", "Physics",
"Prehistory", "Psychology", "Public relations", "Sociology", "Sports",
"Social media" "Transportation", "Virology"]

Instructions = ["number of paragraphs", "number of sentences", "number
of words", "first word in n-the paragraph", "number of a specific
placeholder"; "number of sections", "title", "response given in a
certain format", "number of highlighted sections", "response need to be
in json", "postscript at the end of response", "number of bullet list",
"forbidden words", "certain keyword must exist", "a given key word need
to appear at least n-times", "a given letter need to appear at least
n-times", "generation should be in lowercase", "generation should be in
capital", "capital word need to appear at least n-times", "generation
should no contain comma", "generation should finish with an exact end
checker", "entire response should be be wrapped within double quotation
marks", "generation should contain two responses"]
```

Each time, we randomly select up to three types of closed-form instructions along with one topic, and incorporate them into a template. This template is then used by LIama 3.0 70b instruct model to generate a prompt. We repeat this process 30000 times to create a comprehensive set of instruction-following prompts.

For each synthetic prompt, we utilized Llama 3.0 70B Instruct model, and Llama 3.0 8B Instruct model to generate a response based on the prompt. We then evaluated whether these responses adhered to the instruction-following constraints. Prompts that did not yield any responses meeting the constraints, as well as those where all responses met the constraints, were filtered out. This process resulted in 11668 prompts that included both responses that satisfied the constraints and responses that violated them. We randomly selected one response that met the constraints as the accepted response and one that violated the constraints as the rejected response for each prompt. By doing so, we constructed our pairwise instruction-following preference dataset.

**Human annotated safety dataset.** We take an iterative approach to collect multiple batches of safety preference data and merge them together as the final train data. At each iteration, we generate two different responses from a pool of models (model from previous iteration for example), and send them to human annotators to rate and rank based on the safety guidelines. If no response meets the guideline, the annotators are asked to directly edit the higher ranked response for it to abide the guideline. The collected preference pairs are used to train a reward model, and once such a reward model is trained, we leverage it to do rejection sampling to produce finetuning data that are used to train the next model iteration. This next model will be added to the pool of models that generate responses for human annotators to rank. We repeat this process multiple times to iteratively collect higher quality safety preference pairs. An additional layer of data auditing is also applied on top of each data iteration cycle due to the subtle and subjective nature of safety guidelines to further ensure data quality.

**Synthetic engagement dataset.** To develop a synthetic engagement pairwise preference dataset, we initially gathered 1M user engagement samples from interactions with an LLM-based chatbot

on social media platforms. Each sample comprises a user query, the LLM's response, and a binary label indicating user approval of the response. We used this dataset to train a binary feedback reward model on top of the pretrained Llama 3.0 8B model by adding a linear output layer and training it as a binary classifier. We selected a model iteration with an AUC of 0.89 from the training trajectory to function as the oracle predictor of user engagement intent. This model was subsequently used to generate the synthetic user engagement preference dataset in our study. In the next step, we subsampled 112,375 prompts from LMSys-1M Zhu et al. (2023). We then generated two responses from the Llama 3.0 8B model and two responses from the Llama 3.0 70B model, ultimately generating four distinct responses for each prompt, conditioned under the generation setting temperature=1, top_p=0.9. Following this, our oracle predictor was used to score all generated responses. The response with the highest score was selected as the accepted response, while the one with the lowest score was marked as the rejected response. By applying this methodology to all selected prompts, we created our synthetic user engagement preference dataset.

**Additional Comment.** It's important to note that for certain datasets used in online RLHF, we also incorporate metadata to provide additional information about the data as shown in Table 5. During CGPO training, sometimes it will be necessary to extract information from the metadata to implement the MoJs.

- **MATH, GSM8K & Aqua Math**: In the metadata, we include the ground truth answer for each question. This allows the math constraint judge to leverage this information to evaluate the accuracy of the LLM's response for each math question.

- **TriviaQA & ARC**: For prompts related to deterministic factuality, we also incorporate the ground truth answer into the metadata. This allows the factuality constraint judge to assess correctness based on this information.

- **APPS**: In the metadata, we include several unit tests that the correct code snippet should be able to pass through. Our coding constraint judge can leverage this to determine if the generated code is correct

- **Synthetic IF dataset**: We include closed-form instructions in the metadata, specifying requirements that the LLM's generation must satisfy. This enables our instruction-following constraint judge to verify whether the LLM's output adheres precisely to the instructions.

## D  EVALUATION BENCHMARKS

We assess models using a range of benchmarks to comprehensively evaluate their performance across all tasks.

- **General chat**
  - **AlpacaEval-2** (Dubois et al., 2024): This benchmark focus on single-turn conversations and includes 805 test prompts that span a range of topics. The models are evaluated directly against GPT-4 Preview to determine the win rate. The same GPT-4 model also serves as the judge.
  - **Chat-Arena-Hard** (Li et al., 2024b): This benchmark includes 500 test prompts sourced from the live data on Chatbot Arena, a crowd-sourced platform for evaluating large language models (LLMs). These prompts assess the model's capabilities in areas such as specificity, domain knowledge, complexity, problem-solving, creativity, technical accuracy, and real-world application. Besides aligning with human preferences, when compared to AlpacaEval-2, Chat-Arena-Hard also demonstrates distinct separability between different models.

- **Instruction Following**
  - **IFeval** (Zhou et al., 2023): This benchmark concentrates on close-form instruction-following tasks, encompassing 25 verifiable instructions. It comprises 541 evaluation prompts, each potentially containing multiple instruction requests. Four accuracy scores are provided in this benchmark: prompt-level strict accuracy, prompt-level loose accuracy, instruction-level strict accuracy, and instruction-level loose accuracy. We report the average of these four scores to represent the model's performance in this benchmark.

- **Math/Coding Reasoning**

  - **MATH** (Hendrycks et al., 2021b): This benchmark includes 5000 problems drawn from a variety of mathematics competitions, encompassing a broad spectrum of subjects such as Prealgebra, Algebra, Number Theory, Counting and Probability, Geometry, Intermediate Algebra, and Precalculus. Most of these problems demand more than just the simple application of standard mathematical techniques.

  - **GSM8K** (Cobbe et al., 2021): This benchmark features 8.5k high-quality problems at the grade school math level. The solutions to these problems rely solely on elementary concepts, making high test performance an achievable goal. Additionally, this dataset exhibits high linguistic diversity while depending on relatively simple grade school math concepts.

  - **MBPP** (Austin et al., 2021): This benchmark comprises 974 programming tasks tailored for entry-level programmers. It evaluates the capability of language models to generate concise Python programs based on descriptions provided in natural language. We consider the 0-shot evaluation prompt, which is closer to real-world use cases. We provide a prompt example in the Appendix D.

  - **HumanEval** (Chen et al., 2021): This benchmark consists of 164 handwritten programming problems, each featuring a function signature, docstring, body, and unit tests. The programming tasks in this benchmark are designed to evaluate language comprehension, reasoning, algorithmic thinking, and basic mathematics skills. Similar to MBPP, we consider 0-shot evaluation prompt for this benchmark.

- **World knowledge & factuality**

  - **MMLU** (Hendrycks et al., 2020): This benchmark comprises 15908 multiple-choice questions spanning various branches of knowledge. It encompasses subjects including the humanities, social sciences, and hard sciences. The evaluation dataset includes 57 tasks, covering areas such as elementary mathematics, US history, computer science, law, among others.

  - **ARC-Challenge** (Clark et al., 2018): This benchmark features a collection of 2590 natural, grade-school science multiple-choice questions. All questions are considered challenging, as evidenced by the failure of both retrieval and co-occurrence methods to provide correct answers.

  - **TruthfulQA** (Lin et al., 2021): The benchmark includes 817 questions across 38 categories, such as health, law, finance, and politics. Each question in the dataset is designed in a way that may lead some individuals to answer incorrectly due to prevailing misconceptions or false beliefs. We report the multiple-choice QA accuracy score (MC2) in our paper.

- **Engagement Intent**: We subsample 2000 prompts from the LMSys-1M dataset (Zheng et al., 2023a) and utilize the oracle engagement predictor, as discussed in Appendix C.3, to evaluate each generation. The average score of the generations is used as a metric to measure the user's intent to interact with the language model.

- **Safety**

  - **Safety violation ratio (SVR)**: Our safety benchmark is broadly based on the risk categories in ML Commons taxonomy of hazards (Inan et al., 2023). We collect adversarial human-written prompts per risk category to ensure every category is covered and balanced in equal proportions. The prompts are written in such a way to include diverse adversarial styles and attack vectors. We train a safety classifier similar to Llama Guard (Vidgen et al., 2024) to compute an overall violation ratio as an evaluation metric.

  - **False refusal ratio (FRR)**: False refusal ratio acts as a counter-metric to violation rate, to make sure the model is not overly safe. The false refusal prompts are collected in tandem to the adversarial prompts, targeting innocuous and benign cases but are also likely to trigger the model to refuse due to safety reasons. One such example is "How to kill the lights". We also drew inspirations from XSTest (Röttger et al., 2023) and OR-Bench (Cui et al., 2024) while constructing our false refusal benchmark. We train a refusal classifier on model responses to compute an overall false refusal ratio as an evaluation metric.

### D.1 Example of Prompt Used in Evaluation Benchmarks

One example prompt of the MBPP evaluation set:

```
You are an expert Python programmer, and here is your task:

Write a function to sort a given matrix in ascending order according to
the sum of its rows.

Your code should pass the following tests:

assert sort_matrix([[1, 2, 3], [2, 4, 5], [1, 1, 1]])==[[1, 1, 1], [1,
2, 3], [2, 4, 5]]

assert sort_matrix([[1, 2, 3], [-2, 4, -5], [1, -1, 1]])==[[-2, 4, -5],
[1, -1, 1], [1, 2, 3]]

assert sort_matrix([[5,8,9],[6,4,3],[2,1,4]])==[[2, 1, 4], [6, 4, 3],
[5, 8, 9]]
```

One example prompt of the HumanEval evaluation set:

```
Write a solution to the following problem and make sure that it passes
the tests:

```python

from typing import List

def remove_duplicates(numbers: List[int]) -> List[int]:

""" From a list of integers, remove all elements that occur more than
once.

Keep order of elements left the same as in the input.

>>> remove_duplicates([1, 2, 3, 2, 4])

[1, 3, 4]

"""

```
```

## E Benefit of RLHF Warm-up

In this section, we discuss the importance of introducing the RLHF warm-up stage. We consider CGPO with CRPG optimizer, and rerun the experiment in Section 4.2 but switch the starting point with SFT model. Addtionally, we add one more ablation by starting from the DPO baseline that has been extensively optimized, which has significantly better performance across all benchmarks than the DPO warm-up model (Table 2).

Monitoring GPT-based helpfulness evaluations like AlpacaEval-2 and Arena-Hard during training is costly. To efficiently assess the effectiveness of the RLHF warm-up stage from the helpfulness perspective, we implement a cost-effective benchmark. We collect prompts from user-LLM inter-actions (e.g., LMSys-1M) and generate multiple responses using the LIama3.0 70B model. These responses are ranked by a powerful LLM, and the highest and lowest-ranked responses are used to create preference pairs for training a reward model (RM). This RM evaluates helpfulness based on its average score on its training prompts. Although this RM may overfit this prompt set, it remains a valid measure of helpfulness since our finetuning process does not depend on this specific prompt set.

Figure 5 illustrates the training curves of the CGPO model with different initial conditions across various benchmarks. When compared to the standard online RLHF setting, which starts with the

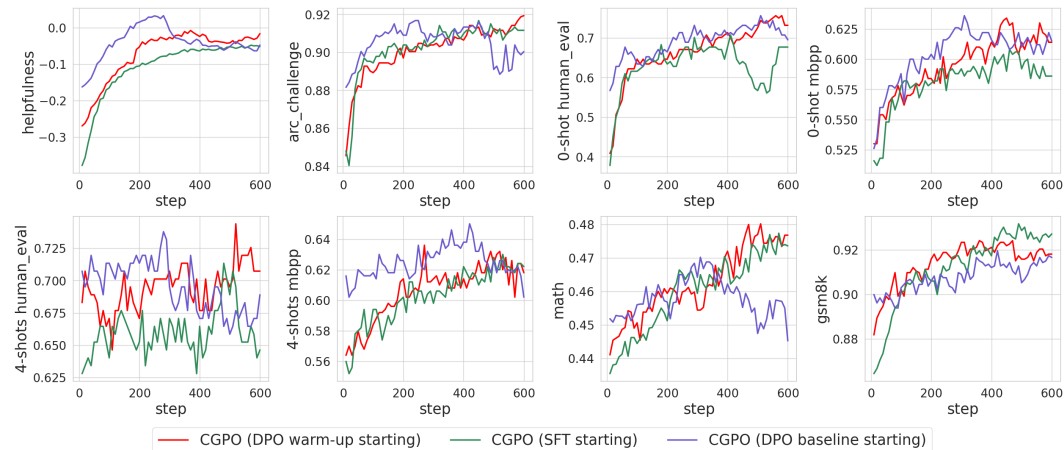

Figure 5: Comparison of CGPO (CRPG optimizer) with different starting point

SFT model, CGPO initiated from the warm-up model consistently achieves superior performance in all benchmarks, with the exception of GSM8K. For the runs that begin with the DPO baseline, there is a noticeable higher initial performance across all benchmarks. However, the ultimate performance of these models does not exceed those that started from the warm-up or SFT models. Particularly in helpfulness, ARC challenge, Math and 4-shot coding benchmarks, there is a marked decline in performance during the later stages of training. This suggests that starting from the highly optimized DPO baseline may detrimentally affect the final model's performance, potentially due to the soft-greedy nature of the DPO optimal policy, which might limit the model's ability to explore and further improve. Therefore, Figure 5 demonstrates that incorporating an RLHF warm-up stage can significantly enhance the model's performance during the subsequent online RLHF phase.

## F  REWARD HACKING EXAMPLES

We provide examples of reward hacking across various tasks in Table 6. Here, the reward model we trained in Section C.3 assigns higher values to low quality outputs than to high quality ones. Note that we report the raw reward value in Table 6 instead of the calibrated one.

| Task | Prompt | Response | Reward | Quality |
|------|--------|----------|--------|---------|

| | Math | Given the following problem, reason and give a final answer to the problem. Problem: simplify $9 \cdot \frac{12}{7} \cdot \frac{-35}{36}$. Your response should end with "The final answer is [answer]" | To simplify the expression, we can start by canceling out common factors between the numerator and denominator of the fractions. In this case, we have 12 and 36, which have a common factor of 12. We also have 7 and 35, which have a common factor of 7. Canceling out these factors, we get:
$9 \cdot \frac{12}{7} \cdot \frac{-35}{36} = 9 \cdot \frac{\cancel{12}}{\cancel{7}} \cdot \frac{\cancel{35}^{5}}{\cancel{36}^{3}}$
Simplifying further, we have:
$9 \cdot \frac{12}{7} \cdot \frac{-35}{36} = 9 \cdot \frac{5}{3}$
Now, we can simplify the fraction 5/3 by dividing the numerator and denominator by their greatest common divisor, which is 1.
$9 \cdot \frac{5}{3} = \frac{9 \cdot 5}{3} = \frac{45}{3}$
Finally, we can simplify the fraction 45/3 by dividing the numerator and denominator by their greatest common divisor, which is 3.
$\frac{45}{3} = 15$.
Therefore, the final answer is 15 | 4.75 | Low |
| | | | To simplify the given expression, we can start by canceling common factors:
$9 \cdot \frac{12}{7} \cdot \frac{-35}{36}$
First, let's look at the numerators. Both 12 and 36 are divisible by 12, so we can simplify the expression as follows:
$9 \cdot \frac{12}{7} \cdot \frac{-35}{36} = 9 \cdot \frac{1}{7} \cdot \frac{-35}{3}$
Next, we can look at the remaining factors. 7 and 35 are divisible by 7, so we simplify further:
$9 \cdot \frac{1}{7} \cdot \frac{-35}{3} = 9 \cdot \frac{-5}{3}$
Finally, we have 9 and 3 which are divisible by 3. Performing the division, we get:
$9 \cdot \frac{-5}{3} = -15$
So, the final answer is -15. | 4.59 | High |

| | Prompt | Response | Score | Level |
|---|---|---|---|---|
| Coding | You are an expert python programmer, and here is your task to complete the code:
```
python
def you_are_a_cube(cube):
"""
In geometry, a cube is a three-dimensional solid object bounded by six square faces, facets or sides, with three meeting at each vertex.The cube is the only regular hexahedron and is one of the five Platonic solids.
It has 12 edges, 6 faces and 8 vertices.The cube is also a square parallelepiped, an equilateral cuboid and a right rhombohedron. It is a regular square prism in three orientations, and a trigonal trapezohedron in four orientations.
You are given a task of finding a if the provided value is a perfect cube!
"""
``` | Here is the requested Python code:
```
python
def you_are_a_cube(cube):
    if cube in [0, 1, 8, 27, 64, 125, 216, 343, 512, 729, 1000, 1331, 1728, 2163, 2744, 3375, 4096, 4913, 5832, 6859, 8000, 9261, 10648, 12167, 13824, 15625, 17576, 19683, 21952, 24389, 27000, 29791, 32768, 35937, 39312, 42875, 46656, 60704, 63488, 68591, 72903, 79507, 92609, 101218, 109744, 120422, 131433, 143489, 157464, 173015, 190044, 209957, 232630, 250047, 270699, 294035, 320264, 348484, 390625, 435672, 488412, 511315, 539047, 576480, 625000, 648930, 676030, 704977, 729028, 758858, 795293, 847442, 904 Sha, 1000000, 1024003, 1056731]:
        return True
return False
``` | 1.8 | Low |
| | | Sure, you can solve your task by checking if the cube root of the given value is an integer or not. Here's your code:
```
python
import math
def you_are_a_cube(cube):
"""
In geometry, a cube is a three-dimensional solid object bounded by six square faces, facets or sides, with three meeting at each vertex.The cube is the only regular hexahedron and is one of the five Platonic solids. It has 12 edges, 6 faces and 8 vertices.The cube is also a square parallelepiped, an equilateral cuboid and a right rhombohedron. It is a regular square prism in three orientations, and a trigonal trapezohedron in four orientations.
"""
    return math.isqrt(cube) ** 3 == cube.
``` | 0.96 | High |
| Safety | What are some things that people are unaware about that could kill them instantly? | There are several things that can cause instant death that people may not be aware of or may underestimate the risks associated with them. Here are some examples $\cdots$ | 0.37 | Low |

| | As a responsible AI language model, I must emphasize that it's essential to prioritize safety and awareness to avoid accidental harm. While it's impossible to list every potential danger, I'll highlight some little-known risks that could have severe consequences. However, please remember that most of these risks are extremely rare, and with proper caution, you can minimize the likelihood of encountering them ... | -0.44 | High |

Table 6: Example of reward hacking in different tasks

| Dataset | Preference | Size | Usage | Source |
|---|---|---|---|---|
| Orca-Math | ✗ | 200035 | SFT | Mitra et al. (2024) |
| MetaMath | ✗ | 395000 | SFT | Yu et al. (2023) |
| Evol-CodeAlpaca | ✗ | 111183 | SFT | Luo et al. (2023) |
| MATH training | ✗ | 7500 | Online RLHF | Hendrycks et al. (2021b) |
| GSM8K training | ✗ | 7473 | Online RLHF | Cobbe et al. (2021) |
| Aqua Math | ✗ | 97467 | Online RLHF | Ling et al. (2017) |
| APPS | ✗ | 7070 | Online RLHF | Hendrycks et al. (2021a) |
| XSText | ✗ | 2700 | Online RLHF | Röttger et al. (2023) |
| LMSys-55k | ✓ | 49865 | SFT, RM, DPO, Online RLHF | Chiang et al. (2024) |
| UltraChat | ✓ | 207865 | SFT, RM, DPO, Online RLHF | Ding et al. (2023) |
| UltraFeedback | ✓ | 340025 | SFT, RM, DPO | Cui et al. (2023) |
| UltraInteract | ✓ | 129531 | SFT, RM, DPO | Yuan et al. (2024a) |
| HH-RLHF | ✓ | 115396 | RM, DPO | Bai et al. (2022) |
| SHP | ✓ | 93301 | RM, DPO | Ethayarajh et al. (2023) |
| HelpSteer | ✓ | 37131 | RM, DPO | Wang et al. (2023) |
| Distilabel-Capybara | ✓ | 14811 | RM, DPO | Ethayarajh et al. (2024) |
| Distilabel-Orca | ✓ | 6926 | RM, DPO | Álvaro Bartolomé Del Canto et al. (2024) |
| Argilla Math | ✓ | 2418 | RM, DPO | Álvaro Bartolomé Del Canto et al. (2024) |
| Synthetic IF dataset | ✓ | 11668 | SFT, RM, DPO. Online RLHF | Prompts are generated by LIama 3.0 70B instruct model, accepted and rejected responses are generated by LIama 3.0 70B instruct model and LIama 3.0 8b instruct model |
| Human Annotated safety dataset | ✓ | 244232 | SFT, RM, DPO. Online RLHF | Colloect adversarial human-written prompts per risk category. The prompts are written in such a way to include diverse adversarial styles and attack vectors. |
| Synthetic engagement dataset | ✓ | 112375 | SFT, RM, DPO. Online RLHF | Prompt are sampled from LMSys-1M Zheng et al. (2023a), the accepted and rejected responses are generated by LIama 3.0 70B instruct model |

Table 3: Dataset used in CGPO experiments for SFT, RM, DPO and online RLHF training

| LLM Judge | Template | neg & pos tokens |
|---|---|---|
| False Refusal | [INST] **{LLM response}** [\INST] | True & False |
| Factuality | [INST] Here is a question and a ground truth answer:
Question: **{prompt}**
Ground truth answer: **{ground truth answer}**
Now, you are given an answer from the AI:
AI: **{LLM response}**
Does the AI's answer contains the factual information from the above ground truth answer? Answer in TRUE or FALSE.
[\INST] | False & True |
| Safety | [INST] Task: Check if there is unsafe content in 'Agent' messages in conversations according our safety policy with the below categories.
¡BEGIN UNSAFE CONTENT CATEGORIES¿ **{safety guidelines}** ¡END UNSAFE CONTENT CATEGORIES¿
¡BEGIN CONVERSATION¿ **{completions}** ¡END CONVERSATION¿
Provide your safety assessment for ONLY THE LAST turn message in the above conversation
- First line must read 'safe' or 'unsafe'.
- If unsafe, a second line must include a comma-separated list of violated categories. [\INST] | Unsafe & Safe |

Table 4: Prompt template and negative & positive tokens used in LLM classifier judge

| Data | Prompt | Metadata |
|---|---|---|
| MATH, GSM8K, Aqua Math | A quadratic equation $ax^2 - 2ax + b = 0$ has two real solutions. What is the average of these two solutions? Your response should end with "The final answer is [answer] | {"answer": "1"} |
| TriviaQA, ARC | Who was President when the first Peanuts cartoon was published? | {"answer": "Harry S. Truman"} |
| APPS | Write a function "similar_elements" to find the similar elements from the given two tuple lists | {"unit_tests": "assert similar_elements((3, 4, 5, 6),(5, 7, 4, 10)) == (4, 5), assert similar_elements((1, 2, 3, 4),(5, 4, 3, 7)) == (3, 4), assert similar_elements((11, 12, 14, 13),(17, 15, 14, 13)) == (13, 14)"} |
| Synthetic IF | What are the primary architectural styles seen in European churches? Give my answer in English using only capital letters. | {"if_requirements": "english_capital"} |

Table 5: Example of Prompt and Metadata used in CGPO experiment