# OpenReview forum: "The Perfect Blend: Redefining RLHF with Mixture of Judges"
_ICLR.cc/2025/Conference — ICLR 2025 Conference Withdrawn Submission_

### Official Review · Reviewer_nYNn · 2024-10-27

**Soundness:** 2
**Presentation:** 2
**Contribution:** 1
**Rating:** 3
**Confidence:** 4

**Summary:**

This paper introduced a simple yet effective approach of multi-objective alignment using constrained optimization.
The method requires some engineering efforts in designing the judges.
The proposed framework is general, and is applicable to PPO / DPO / RAFT.
The authors conducted experiments with all those implementation variants on a variety of tasks.

**Strengths:**

The authors use a variety of tasks to demonstrate their method. They also implemented several variants of the CGPO framework.
The idea is simple, results look promising.

**Weaknesses:**

Some parts of the paper are not easy to follow. In the paragraph begining at line 77, I would recommend the authors to use a Figure 1 to explain their technical contribution rather than the illustrative MoJ, which is very clear - yet the primary-dual constrained optimization part could be further explained with a "figure 1"

Assuming prior knowledge to reward models / how could the hacks be is too strong and lacks a ground.
The paper claims on improving pareto frontier at the intro section but there is no supporting evidence in the experiment section: missing comparison to baseline methods. e.g., [Reward Soups], the authors claimed problems/challenges with this type of methods, yet there is no empirical nor theoretical justification.

Lack of technical novelty. The idea is simple and seems effective, yet applying constrained RL in (multi-objective) alignment has been explored in e.g., [Safe RLHF: Safe Reinforcement Learning from Human Feedback]. This highly-relevant paper is not cited and not compared with.

The introduction of some terms is not necessary, e.g., why would the authors mention the primal-dual forms of constrained optimization? Simplicity could be an advantage if there is no need to make it more complex.

**Questions:**

On the discussion of Vulnerability to Reward Hacking: the claim on the amplification of reward hacking in multi-task learning is not convincing. e.g., the helpfulness can be hacked by generation length, yet the same hack does not work for harmless. Can the authors further justify their claim either theoretically or empirically?

How good is the proposed method when comparing with other multi-objective alignment baselines?

In a multi-task setting, how do the idea differentiates between "objectives" and "constraints"? e.g., in a joint optimization for harmless and helpfulness, which one should be the objective? Will the implementation difficulty of rule-based judge be considered in such a process?

Practically, it is likely that not all of the constraints can be simultaneouly satisfied, how the method tackle this challenge? Will the learning / sample efficiency be extremely low in such cases?

In the experiment section, confidence intervals are only reported in some of the results, is there a specific reason?

---

> ### Author Response · Authors · 2024-11-22
> **We appreciate the reviewer's comments and feedback. While we respectfully disagree with a few points raised, we value the opportunity to address them. Below are our detailed responses**
>
> $\textbf{Q1}$: Assuming prior knowledge to reward models / how could the hacks be is too strong and lacks a ground.
>
> $\textbf{A1}$: $\textbf{We respectfully disagree with the reviewer's assertion}$. The vulnerabilities of reward models and the patterns of reward hacking can be effectively identified. For instance, we can analyze the failure modes of the reward model using a hold-out dataset. Additionally, by running policy optimization with a reward model over sufficiently long steps, we can employ an LLM-based judge to detect reward hacking patterns in the resulting policy. Furthermore, in practical real-world applications, LLM are typically updated iteratively. As a result, it is not uncommon for reward hacking patterns to be fully captured and addressed in previous iterations.
>
> $\textbf{Q2}$: The paper claims on improving pareto frontier at the intro section but there is no supporting evidence in the experiment section: missing comparison to baseline methods. e.g., [Reward Soups].
>
> $\textbf{A2}$: $\textbf{We respectfully disagree with the reviewer's assertion}$. The Reward Soup approach is not directly comparable to our work due to significant differences in the scope of objectives considered. In our study, we address eight objectives: general helpfulness, instruction following, math, coding, factuality, user engagement, false refusal, and safety. In contrast, the original Reward Soup paper considers at most four objectives. While it is theoretically possible to extend Reward Soup to accommodate more objectives, neither the original paper nor existing literature provides empirical evidence supporting its scalability to an eight-objective setting. In practice, scaling Reward Soup to handle eight objectives is both costly and challenging, warranting a separate, dedicated study. Given the absence of such empirical studies, we believe it is not appropriate to use RewardSoup as a baseline in our context.
>
> $\textbf{Q3}$: Lack of technical novelty. The idea is simple and seems effective, yet applying constrained RL in (multi-objective) alignment has been explored in e.g., [Safe RLHF: Safe Reinforcement Learning from Human Feedback]. This highly-relevant paper is not cited and not compared with.
>
> $\textbf{A3}$: $\textbf{We respectfully disagree with the reviewer's assertion}$. Our method introduces novel contributions and demonstrates clear advantages over previous work:
>
> - The approach in the Safe RLHF paper utilizes a primal-dual method, relying on the standard primal-dual constrained RL optimizer. In contrast, we introduce three new primal-type constrained optimizers, which have not been explored in any previous works before.
> Our proposed primal-type RLHF method offers significant advantages over the primal-dual approach used in the Safe RLHF paper. Specifically, our method scales more effectively to multiple constraints and objective settings. The safety RLHF approach, on the other hand, struggles with scalability and is highly sensitive to optimization hyperparameters, making it challenging to apply in the context of LLM. These limitations of primal-dual approach have been extensively discussed in traditional constrained RL literature.
>
> - Furthermore, the Safe RLHF method is not applicable to our setting, as we consider a more general framework where constraints are not necessarily differentiable RL objectives. In contrast, the Safe RLHF paper formulates safety constraints as differentiable RL objective. Our CGPO method is applicable to a broader range of scenarios where the Safe RLHF approach is not suitable.
>
> - The primal-dual approach in the Safe RLHF paper is not directly applicable to our work due to significant differences in the number of objectives and constraints. In our experiments, we consider three reward models and six constraints, whereas the Safe RLHF paper addresses only one objective and one constraint. While it is theoretically possible to extend the Safe RLHF method to accommodate more reward models and constraints, neither the original paper nor existing literature provides empirical evidence supporting its scalability to our setting. Given the absence of such empirical studies, we believe it is not appropriate to use the Safe RLHF method as a baseline in our context.

---

> ### Author Response · Authors · 2024-11-22
>
> $\textbf{Q4}$: The introduction of some terms is not necessary, e.g., why would the authors mention the primal-dual forms of constrained optimization? Simplicity could be an advantage if there is no need to make it more complex
>
> $\textbf{A4}$: We respectfully disagree with the reviewer's comment.  We believe it is important to highlight the distinctions between our approach and previous works to clearly convey its novelty to readers. The introduction of the primal-type optimizer is a key feature of our method, offering significant advantages over traditional primal-dual methods, particularly in settings with multiple constraints and objectives. By emphasizing this aspect, we aim to underscore the unique contributions and effectiveness of our approach
>
> $\textbf{Q5}$: On the discussion of Vulnerability to Reward Hacking: the claim on the amplification of reward hacking in multi-task learning is not convincing. e.g., the helpfulness can be hacked by generation length, yet the same hack does not work for harmless. Can the authors further justify their claim either theoretically or empirically?
>
> $\textbf{A5}$: We appreciate the reviewer's feedback on this point. To clarify, our intention in the introduction was to emphasize that resolving reward hacking issues is more challenging in a multi-task setting when using previous methods, such as early stopping, compared to addressing these issues in a single-task setting. We will ensure this is more clearly illustrated in our revision.
>
>
> $\textbf{Q6}$: How good is the proposed method when comparing with other multi-objective alignment baselines?
>
> $\textbf{A6}$: As previously mentioned, it is unclear whether the RewardSoup and Safe RLHF approaches are applicable to the scope of the problems we address. Due to the absence of empirical studies demonstrating the feasibility of these methods in our setting, we believe it is not appropriate to use either RewardSoup or Safe RLHF as baselines for comparison in our paper.
>
>
> $\textbf{Q7}$: In a multi-task setting, how do the ideas differentiate between "objectives" and "constraints"? e.g., in a joint optimization for harmless and helpfulness, which one should be the objective? Will the implementation difficulty of a rule-based judge be considered in such a process?
>
> $\textbf{A7}$: In previous works, such as Safe RLHF, both helpfulness and safety reward models are applied to the same set of prompts, necessitating that one be treated as the objective and the other as a constraint. However, in our CGPO framework, we adopt a different approach. We apply the helpfulness and safety reward models to two disjoint sets of prompts, allowing both reward models to be treated as objectives. This enables us to optimize both rewards simultaneously on their respective prompts.
>
> The distinction between objectives and constraints is not influenced by the difficulty of implementing a rule-based judge. In all tasks we consider, there is a preference-based reward model, which serves as the objective. Rule-based judges are used to complement this preference-based reward model, rather than determining the classification of objectives and constraints.

---

> > ### Author Response · Authors · 2024-11-22
> >
> > $\textbf{Q8}$: Practically, it is likely that not all of the constraints can be simultaneously satisfied, how does the method tackle this challenge? Will the learning / sample efficiency be extremely low in such cases?
> >
> > $\textbf{A8}$: The primary purpose of introducing constraints in CGPO is to address the limitations of the reward model. Our goal with CGPO is to mitigate reward hacking and enhance RLHF alignment results compared to baseline RLHF methods. This differs from the goal of traditional constrained optimization, which aims to fully satisfy all constraints. Therefore, even if not all constraints can be fully satisfied, our approach remains effective in improving the alignment results.
> > Unlike traditional constraint methods, such as the primal-dual approach, which may zig-zag between feasible regions of different constraints and result in low sample efficiency, CGPO employs primal-type approaches. These approaches focus on avoiding the maximization of the likelihood of violating samples, rather than enforcing the satisfaction of all constraints. As a result, CGPO does not experience the same zig-zag issue as traditional primal-dual methods and thus maintains better sample efficiency.
> >
> >
> > $\textbf{Q9}$: In the experiment section, confidence intervals are only reported in some of the results, is there a specific reason?
> >
> > $\textbf{A9}$: For all benchmarks, except AlpacaEval-2 and Arena-Hard, we employ greedy decoding during inference, which eliminates randomness in our evaluation process. Consequently, confidence intervals are not necessary for these results. Additionally, it's important to note that evaluating a 70B scale LLM incurs significant computational costs. It is quite rare in the research community to evaluate a model's performance throughout the entire training trajectory across multiple benchmarks, as we have done in our study. Therefore, we believe our evaluation is both comprehensive and robust.
> >
> > Thank you once again for your valuable feedback. We sincerely hope that our responses have adequately addressed your concerns. If you find them satisfactory, we kindly request you to consider increasing your evaluation rating. Additionally, we are more than willing to address any further questions or concerns you may have. Thank you for your thoughtful review and feedback.

---

> > > ### Comment · Reviewer_nYNn · 2024-11-29
> > > **Thanks for the response**
> > >
> > > Thank you for the responses. However, I do not find the responses to be very informative nor convincing.
> > >
> > > In Q2 and Q6, I acknowledge that implementing all eight objectives might be challenging in your setup. However, the baselines mentioned are clearly relevant and important for a fair comparison. As authors, it is your responsibility to address these comparisons rather than avoid them. If implementing all objectives is too burdensome, perhaps comparing under a subset of four objectives could strike a balance between feasibility and thoroughness. I don’t see why such comparisons wouldn’t be possible.
> > >
> > > For Q3, I don’t find it constructive to frame your response as “disagreeing with the reviewer” on a missing citation. The absence of an important reference is a factual matter, right?
> > >
> > > In Q8, my question was not answered directly.
> > >
> > > In future revisions, I would recommend the authors make fair comparisons with existing baselines. **Hard to implement** should never be a reason for missing comparisons.

---

> ### Author Response · Authors · 2024-11-29
>
> Thank you for your feedback and for taking the time to review our work
>
> Regarding Q2 and Q6:
> We understand the importance of fair comparisons with existing baselines. However, our paper focuses on the challenging real-world problem of large-scale LLM post-training with multiple objectives and constraints, and the $\textbf{goal of our work is developing a method to solve the problem with the scope that previous methods are not able to solve at all}$, instead of competing with previous methods in small scale problems. This scope necessitates comparisons with methods that share similar scalability, such as DPO and PPO. $\textbf{We respectfully maintain that "reward souping" and "safeRLHF" are not an appropriate baselines}$ for our study, given the specific nature and scale of the problem we address.
>
> Regarding Q3:
> Our intention was to emphasis that $\textbf{we are not agree with reviewer's claim that our work lacks novelty}$. The assertion that our work is not novel because constrained RL methods have been considered in other alignment literature is $\textbf{inappropriate}$. While we acknowledge the relevance of Safe RLHF and will include it in our revision, we believe our contributions are clearly distinct. Specifically:
> - Unlike previous works that adopt existing primal-dual constrained policy optimization methods, we introduce three novel primal-type policy optimization methods. These methods are designed to scale effectively to large-scale settings with numerous objectives and constraints, which is a significant advancement over existing approaches that have been demonstrated only in smaller-scale problems.
> - Our work addresses a more general problem where constraints can be non-differentiable, a scenario that previous methods cannot handle.

---

> ### Author Response · Authors · 2024-11-29
>
> Regarding Q8: To clarify further, the introduction of constraints in our approach is intended to compensate for the limitations of the reward model, rather than to enforce strict adherence to all constraints, which may exceed the model's capabilities and cause unnecessary sample inefficiency. We utilize a primal-type optimizer to ensure that our method remains robust even when some constraints are not fully satisfied. This approach helps us avoid issues related to sample inefficiency. Consequently, our method achieves efficiency comparable to non-constraint optimizers. While some constraints may be violated in the end, our approach still demonstrates significantly better performance and a lower constraint violation ratio compared to the unconstrained baseline method.

---

### Official Review · Reviewer_JaGE · 2024-11-02

**Soundness:** 3
**Presentation:** 4
**Contribution:** 3
**Rating:** 6
**Confidence:** 4

**Summary:**

This work proposes a paradigm to align LLMs in constrained multi-objective settings. It first introduces the notion of ``mixture of judges’’, which utilizes rules or LLMs to check if a given response satisfies a set of constraints. Then, this work utilizes CGDO to optimize the reward while complying to the constraints. To extend the algorithm to the multi-objective setting, this work proposes CRPG, CRRAFT, CODPO that utilize multiple reward models and integrate the gradient updates.

**Strengths:**

- The manuscript is well-written; and the proposed algorithm is straightforward to understand.
- This work considers the constrained multi-objective alignment setting, which is novel and important. Also, compared to previous works which usually specify probabilistic constraints, this work specifies a stricter constraint, i.e., $P_{s \sim \mathcal{D}, a\sim\pi_{w}}((s,a) \in \Sigma) \geq 1$.
- According to Table 2, the proposed algorithm(s) outperform their DPO, PPO baselines.

**Weaknesses:**

- While the work studies a constrained optimization problem, the constrain satisfaction in Equation (3) is not verified.
- In line 109, this work claims that the proposed method could avoid compromises due to conflicting goals from other tasks; however, in Algorithm 2, in the parameter updating step (step 6), conflicting goals might induce conflicting gradients ($\tilde g_l(\pi_{w_{t}})$), which stills leads to compromises.
- Since the MoJ is involved in the training process, the cost of LLM calling could be high. (As the practice in several previous works like [1], it is possible to model the model the constrains using collected safety data. This avoids calling LLMs during the training process. )

[1] Dai, etc. Safe RLHF: Safe Reinforcement Learning from Human Feedback.

**Questions:**

- What are the LLM models used as the LLM judges?
- What are FRR and SVR in Table 2?

---

> ### Author Response · Authors · 2024-11-22
> **We thank the reviewer for the valuable comments and feedback, the following are our responses**
>
> $\textbf{Q1}$: While the work studies a constrained optimization problem, the constraint satisfaction in Equation (3) is not verified.
>
> $\textbf{A1}$:The primary purpose of introducing constraints in CGPO is to address the limitations of the reward model. Our goal with CGPO is to mitigate reward hacking and improve RLHF alignment results compared to baseline RLHF methods, which differs from the goal of traditional constrained optimization methods to fully satisfy the constraints.
>
> Empirically, we observed that during training, certain constraints such as factuality, false refusal, and safety see their violation ratios decrease and converge to nearly zero. However, for other constraints like math, coding, and instruction following, the violation ratios significantly decrease from their initial values but do not converge to zero. This is due to the limited expressive power of the model. We will include a new section in the appendix to summarize these observations during training.
>
> $\textbf{Q2}$: In line 109, this work claims that the proposed method could avoid compromises due to conflicting goals from other tasks; however, in Algorithm 2, in the parameter updating step (step 6), conflicting goals might induce conflicting gradients, which still leads to compromises.
>
> $\textbf{A2}$: Thank you for your insightful question. In traditional multi-objective RLHF methods, reward models are typically combined linearly, as shown in the formulation:
>
> $\mathcal{L}_{\text{linear}}(\pi) = \mathbb{E}\left[ \sum_i w_iR_i(x), \pi \right]$, where $x \in \Omega_1 \cup \Omega_2 \cup \cdots$
>
> In this approach, all reward models share the same domain, which can lead to compromises when conflicting goals are present, as the gradients from different objectives are combined.
>
> In contrast, our method, CGPO, explicitly assigns disjoint domains to different reward models. The formulation is as follows:
>
> $\mathcal{L}_{\text{CGPO}}(\pi) = \mathbb{E}[R(x), \pi]$
>
> where $R(x) = R_1(x)$ when $x\in\Omega_1$, $R(x) = R_2(x)$ when $x\in\Omega_2$, ...
>
> By ensuring that each reward model operates within its own domain, CGPO effectively prevents conflicting goals from being applied to the same data. This separation allows us to mitigate compromises that arise from conflicting gradients, as each reward model influences only its designated subset of data. Consequently, CGPO can better maintain the integrity of each objective without the interference that occurs in linear combination methods.
>
> We hope this explanation clarifies how CGPO addresses the issue of conflicting goals and reduces the need for compromises.
>
> $\textbf{Q3}$: Since the MoJ is involved in the training process, the cost of LLM calling could be high. (As the practice in several previous works like [1], it is possible to model the constraints using collected safety data. This avoids calling LLMs during the training process. )
>
> $\textbf{A3}$: In CGPO all LLM judges only do one-token inference (simply output True or False for constraint assessment), thus the cost of LLM calling is cheap and equivalent to using a binary classifier as [1].
>
> $\textbf{Q4}$: What are the LLM models used as the LLM judges?
>
> $\textbf{A4}$: For factuality judge we reuse the policy model by itself, developed based on llama 3.0 70B
>
> For safety judge, false refusal judge they are developed based on llama 3.0 8B.
>
> All other judges are rule-based and do not involve LLMs.
> For details of all  judges please refer to our appendix.
>
> $\textbf{Q5}$: What are FRR and SVR in Table 2?
>
> $\textbf{A5}$: FRR represents false-refusal-rate and SVR represents safety-violation-rate, we will add more context in our revision.
>
> Thank you once again for your valuable feedback. We sincerely hope that our responses have adequately addressed your concerns. If you find them satisfactory, we kindly request you to consider increasing your evaluation rating. Additionally, we are more than willing to address any further questions or concerns you may have. Thank you for your thoughtful review and feedback.

---

> ### Author Response · Authors · 2024-11-24
>
> We hope this message finds you well. We kindly want to check if you had a chance to review our rebuttal, and if you have any further questions or comments we can address to help with your evaluation. Thanks again for your efforts and suggestions in improving our manuscript.

---

### Official Review · Reviewer_EeBp · 2024-11-06

**Soundness:** 3
**Presentation:** 3
**Contribution:** 3
**Rating:** 6
**Confidence:** 3

**Summary:**

This paper introduces a novel training paradigm called Constrained Generative Policy Optimization (CGPO), to tackle the challenge of multi-objective optimization and multi-constraints in RLHF. Under this new training paradigm there are several critical building blocks; the authors developed constrained RLHF optimizers, judges for handling the constraints, a new strategy of performing RLHF with the consideration of different types of data, reward models, and judges. The authors instantiated the proposed framework on Lllama3.0 and reported highly effective results compared to the baselines.

**Strengths:**

1. The authors proposed a new training paradigm for RLHF which handles multi-objective and multi-constraints, which can address several limitations of existing paradigms. The proposed paradigm is highly modular and has the potential to scale to more complicated future scenarios.
2. The experimental design is thoughtful, spanning a comprehensive set of tasks and constraints. The authors performed extensive evaluations and demonstrated the effectiveness of CGPO.

**Weaknesses:**

1. I appreciate the thoughtful design of the multi-objective and the multi-constraints, as well as the customizability of the framework. But there are a few natural questions following these:

    a) For all these "customized combinations" (line 292), "tailor the specific reward model to be applied for each task" (line 336), "uniquely tailored for each task" (line 347), "specifically tailored hyperparameter setup" (line 353) -- it's unclear how these design choices are made, and if one can reproduce these procedures, and how to use this framework for future needs. For the purpose of not only reproducibility but also the extensibility to new data/needs that may arise in the future, it's critical to: 1) document the procedures of making each specific design choice; 2) provide general guidelines that one can follow to develop these components; 3) discuss potential ways of automating some design choices.

    b) Given the complexity of the framework, it would be nice to 1) understand the contribution/sensitivity of each sub-component (with some ablation), 2) understand the complexity overhead.

2. In terms of addressing the shortcoming of using a linear combination of reward models (a.k.a. linear scalarization) on all data, the authors proposed an approach that selectively applies compatible reward models to selective sets of data, which however still falls within the regime of linear scalarization types of methods. The literature [1] has pointed out that linear scalarization is incapable of fully exploring the Pareto frontier, while specialized multi-task optimizers, e.g. MGDA, can find more balanced solutions. It's worth considering the feasibility of applying MGDA and discussing the trade-off in complexity and performance.

**Minors**

- Typos: Line 181: optimizaiton; sec 3.2 header Multi-Taks.


**References**

[1] Hu, Yuzheng, et al. "Revisiting scalarization in multi-task learning: A theoretical perspective." *Advances in Neural Information Processing Systems* 36 (2024).

**Questions:**

See "weaknesses" section; would appreciate clarifications/discussions from the authors.

---

> ### Author Response · Authors · 2024-11-22
> **We thank the reviewer for the valuable comments and feedback, the following are our responses.**
>
> $\textbf{Q1}$: For the purpose of not only reproducibility but also the extensibility to new data/needs that may arise in the future, it's critical to: 1) document the procedures of making each specific design choice; 2) provide general guidelines that one can follow to develop these components; 3) discuss potential ways of automating some design choices.
>
> $\textbf{A1}$: We have outlined the procedures for making customized design choices as follows:
>
> Part I: Customized Reward Model
> Guideline: Select a reward model that is specifically trained with task-relevant preference data for each task.
>
> Example: For safety tasks, use reward models trained with harmfulness preference data. For general chat tasks, use models trained with helpfulness and honesty preference data.
>
> Part II: Customized Judge
> Guideline: Choose a judge to address the weaknesses of the reward model used in each task.
>
> Evaluation: Assess the reward model's weaknesses using a holdout dataset. Identify key failure modes from data points where the model fails to predict preferences accurately.
>
> Selection: Based on this analysis, select a judge. This process can involve human-in-the-loop evaluation or leveraging LLMs to assess according to human-created rules.
>
> Part III: Customized Optimizer Hyperparameters
> Guideline: The choice of optimizer hyperparameters should consider the reward model's quality and the problem's nature.
>
> High-Accuracy Models: For tasks with highly accurate reward models, adjust regularization/constraints to allow the policy to converge to larger KL values.
>
> Exploratory Tasks: For tasks like math and coding that benefit from exploration, use larger generation numbers per prompt.
>
> Automation of Design Choices
>
> Parts I & II: Automation is feasible by utilizing LLMs for data partitioning and failure mode analysis.
>
> Part III: May still require human expertise and inspection to prioritize knowledge and ensure optimal settings.
>
> We will add a new section to include all above content in our revision.
>
> $\textbf{Q2}$: Whether CGPO still falls within the regime of linear scalarization types of methods, and whether MGDA in literature [1] can be applied.
>
> $\textbf{A2}$: Thank you for your insightful comments and the reference provided. We appreciate the opportunity to clarify our approach.
>
> The multi-task reward modeling method in CGPO does not align with the linear scalarization approach as defined in [1]. We illustrate this distinction below:
>
> The CGPO loss function can be interpreted as an RL loss function with a single reward model that operates with piece-wise functionality:
>
> $\mathcal{L}_{\text{CGPO}}(\pi) = \mathbb{E}[R(x), \pi]$
>
> where $R(x) = R_1(x)$ when $x\in\Omega_1$, $R(x) = R_2(x)$ when $x\in\Omega_2$, ...
>
> In contrast, the linear scalarization loss function is defined as:
>
> $\mathcal{L}_{\text{linear}}(\pi) = \mathbb{E}\left[ \sum_i w_iR_i(x), \pi \right]$, where $x \in \Omega_1 \cup \Omega_2 \cup \cdots$
>
> As demonstrated, these two loss functions are fundamentally different.
>
> Furthermore, the problem addressed by CGPO differs from traditional multi-objective optimization. In CGPO, the domains of different reward models are disjoint, whereas traditional multi-objective methods, as discussed in [1], assume shared domains for all reward models. This fundamental difference means that MGDA and CGPO represent two distinct directions, and MGDA is not directly applicable to the problem considered by CGPO.
>
> However, it would be intriguing to explore the application of MGDA within a single task by decomposing the original single reward model into different dimensions. By doing so, we could create sub-reward models within a single task that share the same domain, allowing us to apply MGDA techniques.
>
> Thank you once again for your valuable feedback. We sincerely hope that our responses have adequately addressed your concerns. If you find them satisfactory, we kindly request you to consider increasing your evaluation rating. Additionally, we are more than willing to address any further questions or concerns you may have. Thank you for your thoughtful review and feedback.

---

> ### Author Response · Authors · 2024-11-24
>
> We hope this message finds you well. We kindly want to check if you had a chance to review our rebuttal, and if you have any further questions or comments we can address to help with your evaluation. Thanks again for your efforts and suggestions in improving our manuscript.

---

### Note · Authors · 2025-01-22

I have read and agree with the venue's withdrawal policy on behalf of myself and my co-authors.